# GMSA: Enhancing Context Compression via Group Merging and Layer Semantic Alignment

## Abstract

Large language models (LLMs) have achieved impressive performance in a variety
of natural language processing (NLP) tasks. However, when applied to long-
context scenarios, they face two challenges, i.e., low computational efficiency and
much redundant information. This paper introduces GMSA, a context compression
framework based on the encoder-decoder architecture, which addresses these chal-
lenges by reducing input sequence length and redundant information. Structurally,
GMSA has two key components: **Group Merging** and **Layer Semantic Align-
ment (LSA)**. Group merging is used to effectively and efficiently extract summary
vectors from the original context. Layer semantic alignment, on the other hand,
aligns the high-level summary vectors with the low-level primary input semantics,
thus bridging the semantic gap between different layers. In the training process,
GMSA first learns soft tokens that contain complete semantics through autoencoder
training. To furtherly adapt GMSA to downstream tasks, we propose **Knowledge
Extraction Fine-tuning (KEFT)** to extract knowledge from the soft tokens for
downstream tasks. We train GMSA by randomly sampling the compression rate
for each sample in the dataset. Under this condition, GMSA not only significantly
outperforms the traditional compression paradigm in context restoration but also
achieves stable and significantly faster convergence with only a few encoder layers.
In downstream question-answering (QA) tasks, GMSA can achieve approximately
a 2x speedup in end-to-end inference while outperforming both the original input
prompts and various state-of-the-art (SOTA) methods by a large margin. [1]

## 1  Introduction

Thanks to powerful reasoning and generalization capabilities, large language models (LLMs) have
achieved remarkable performance across various natural language processing (NLP) tasks [23, 26].
However, directly applying LLMs to long-context scenarios presents two challenges: (1) Compu-
tational efficiency. When processing long prompts, the quadratic complexity of the Transformer's
attention mechanism [27] results in long inference latency. (2) Redundant information. Much
redundant information in long-context scenarios can degrade model performance [10].

Prompt compression methods address these two challenges by significantly reducing input length
and removing redundant information. Prompt compression can be categorized into hard prompt
compression [14, 11, 21, 10, 25, 38, 3, 4, 37] and soft prompt compression [20, 6, 7, 34, 16]. Hard
prompt compression involves deleting certain tokens from the original context to achieve compression.
However, this explicit compression approach inevitably compromises semantic integrity. In contrast,
leveraging the inherent redundancy in semantic vectors, soft prompt compression learns a set of soft

---

[1]Our code and models will be released after acceptance.

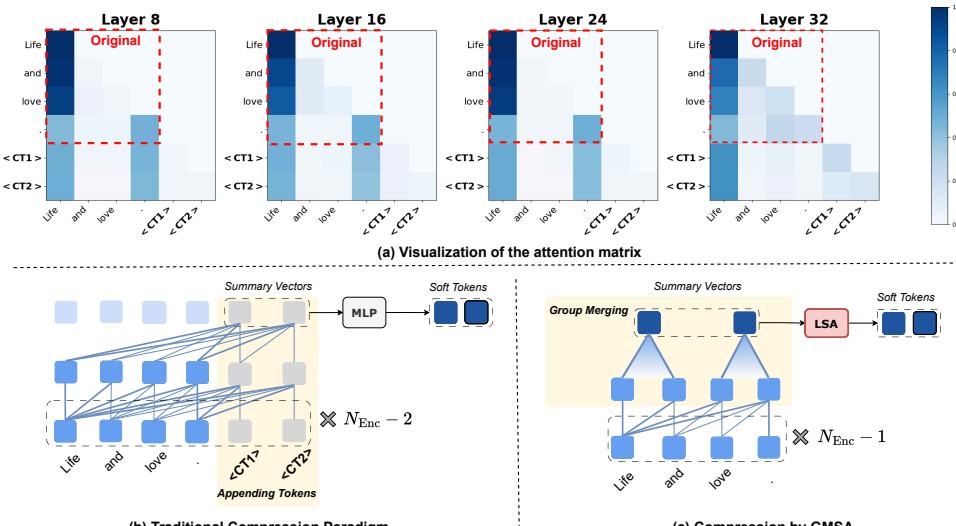

(a) Visualization of the attention matrix

(b) Traditional Compression Paradigm

(c) Compression by GMSA

Figure 1: **Traditional Compression Paradigm v.s. Compression by GMSA.** (a) visualizes the attention matrix when processing *Life and love. <CT1> <CT2>*, where *<CT1>* and *<CT2>* are randomly initialized tokens. *Original* shows the attention changes during processing of *life and love.* (b) represents the traditional compression paradigm. It first learns summary vectors in an autoregressive manner layer by layer, and then completes coarse-grained semantic alignment through a multi-layer perceptron (MLP), where $N_{\mathrm{Enc}}$ denotes the number of encoder layers. (c) denotes the compression paradigm of GMSA, which first learns summary vectors through group merging and completes semantic alignment between different layers through the Layer Semantic Alignment (LSA) module.

tokens with a length much shorter than the original context, enabling compression while preserving complete semantics.

Although existing soft prompt compression methods can effectively reduce the number of input tokens, they have two limitations: (1) Uneven semantics learning and non-parallelism. As shown in Figure 1, in the traditional compression paradigm, the appending randomly initialized tokens learn summary vectors layer by layer in an autoregressive manner. LLM tends to aggregate information on a few anchor tokens [36, 29, 9, 28]. The semantics of anchor tokens (i.e., *Life* and .) are emphasized layer by layer, resulting in the semantics of the summary vectors being dominated by them while the semantics of other tokens are diluted (i.e., uneven semantics learning). This limits the retention of complete semantics, and the process cannot be parallelized. (2) Ignoring the significant gap between the semantic representations of different layers in LLMs [12, 19]. The summary vectors, which represent high-level semantics and are highly abstract, are directly treated as ordinary tokens (i.e., as low-level semantic information) and input into the decoder during training and testing, resulting in a large semantic gap. Therefore, two questions naturally arise: (1) *How can we effectively and efficiently learn summary vectors?* (2) *How can we bridge the significant semantic gap between different layers?*

To address the aforementioned limitations, we propose GMSA, a context compression framework based on the encoder-decoder architecture, which innovatively resolves these limitations from a structural perspective. Specifically, we tackle the first limitation through **Group Merging**. Group merging performs grouping and merging on the last hidden state of the encoder (see Figure 1). To avoid dilution and achieve more uniform semantic learning, group merging equally considers each group, and within each group, all tokens are merged via averaging pooling. Group merging is conducive to retaining complete semantic information and supports parallelization for *effective and efficient* semantic extraction. Subsequently, to address the second limitation, we bridge the gap between highly abstract high-level semantic information and low-level primary input semantics by passing the summary vectors through the **Layer Semantic Alignment (LSA)** module, which is composed of several Transformer blocks initialized with the weights of lower-layer decoder blocks (see Figure 2). *This step allows the summary vectors containing high-level abstract semantic*

*information to be mapped into a low-level semantic space, thereby bridging the semantic gap between different layers.*

During the training process, GMSA first employs the autoencoder training to ensure that the generated soft tokens contain complete semantic representations. Building on this foundation, we further propose **Knowledge Extraction Fine-tuning (KEFT)** to adapt GMSA to downstream tasks. Specifically, we freeze the encoder and LSA (which, after autoencoder training, can already produce soft tokens containing complete semantics) and fine-tune the decoder to enhance its ability to extract knowledge from the soft tokens.

Our contributions are threefold: (1) Structurally, we introduce the GMSA, which effectively and efficiently learns summary vectors through group merging and bridges the semantic gap between different layers via a layer semantic alignment (LSA) module. (2) In the training process, we propose Knowledge Extraction Fine-tuning (KEFT) to guide the decoder to extract the knowledge required by downstream tasks from soft tokens. (3) Experimental results in context restoration and multi-document question answering demonstrate the effectiveness and superiority of our method, e.g., on NaturalQuestions with an 8x compression constraint, GMSA achieves approximately 36% higher Exact Match compared to the original input prompt, while also realizing a 2x end-to-end speedup.

## 2 Problem Formulation

Given a retrieval-augmented prompt $X = (X^{\text{ins}}, X^{d_1}, ..., X^{d_k}, ..., X^{d_K}, X^{\text{q}})$, where $X^{ins}$, $\{X^{d_k}\}_{k=1}^{K}$, and $X^{\text{q}}$ represent the instruction, context, and input question respectively. The prompt has a total token length $L$. The key aspect of the context compression system lies in generating a compressed prompt $\widetilde{X}$ with length $\widetilde{L}$, where the compression rate is defined as $\tau = \frac{L}{\widetilde{L}}$. Let $y$ denote the ground truth answer given the original input $X$, and $\widetilde{y}$ denote the answer generated by the large language model (LLM) when input with the compressed prompt $\widetilde{x}$. We aim for the distributions of $y$ and $\widetilde{y}$ to be similar under high compression rates $\tau$. This can be formulated as:

$$\min_{\widetilde{x}, \tau} \text{KL} \left( P\left( \widetilde{y} \mid \widetilde{X} \right), P\left( y \mid X \right) \right) \tag{1}$$

Due to space limitations, we introduce related work in Appendix A.

## 3 Method

### 3.1 GMSA Architecture

In this section, we elaborate on the architecture of our proposed context compression framework, GMSA, which includes two key components: group merging and layer semantic alignment (LSA). GMSA undergoes a two-stage training process: autoencoder training (see Figure 2) and knowledge extraction fine-tuning (KEFT) (see Figure 3). First, GMSA ensures that the generated soft tokens contain the complete semantic representation of the original text through the autoencoder training process. Then, it applies the knowledge contained in the soft tokens to downstream tasks via KEFT.

### 3.2 Group Merging

**Extraction of Semantic Features.** First, we extract the semantic features of the original text through a $k$-layer language modeling model as the encoder. The encoder is trained using LoRA.

$$H = \text{Encoder}_k(X), \tag{2}$$

where $X$ is the original text and $H$ is the obtained last hidden state.

**Merging.** We divide the obtained $H$ into several groups according to the size of the compression limit, as the group length $L_G$ (e.g., when the compression rate is 4, the group length is also 4). To this end, original text representations are organized as follows:

$$H = \left[ H_1, \ldots, H_{\mathbf{G}_j}, \ldots, H_{\mathbf{G}_{N_g}} \right]$$
$$= \left[ H_{1:L_G}, \ldots, H_{(j-1) \times L_G : j \times L_G}, \ldots, H_{N_d - L_G + 1 : N_d} \right].$$

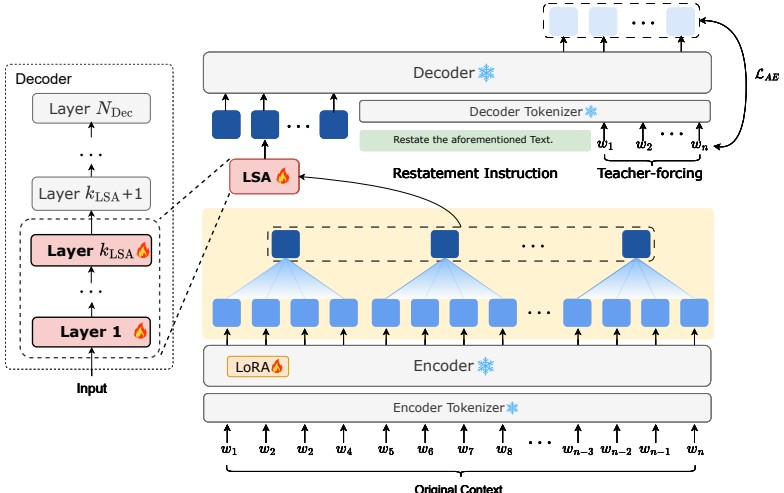

Figure 2: **The Autoencoder Training Process of GMSA.** GMSA consists of an encoder and a decoder, trained in an autoencoder manner using cross-entropy loss. GMSA first generates a set of summary vectors that meet the compression rate by performing group merging on the last hidden state of the encoder, and then achieves cross-layer semantic alignment through the Layer Semantic Alignment (LSA) module, which is composed of several Transformer blocks initialized with the weights of lower-layer decoder blocks. Remarkably, we find that using just a single layer of LSA can achieve excellent semantic preservation (see Appendix B), hence $k_{\text{LSA}} << N_{\text{Dec}}$.

We take the average of each dimension of each group token to obtain the initial compressed representation.

$$\widetilde{H} = \left[\bar{H}_{\mathbf{G}_1}, ..., \bar{H}_{\mathbf{G}_i}, ..., \bar{H}_{\mathbf{G}_N}\right]$$
$$= \left[\frac{1}{L_G}\sum H_{\mathbf{G}_1}, ..., \frac{1}{L_G}\sum H_{\mathbf{G}_i}, ..., \frac{1}{L_G}\sum H_{\mathbf{G}_N}\right]$$

, where $\widetilde{H}$ is the obtained initial compressed representation.

### 3.3 Layer Semantic Alignment

The layer semantic alignment (LSA) module is used to complete the alignment from the soft tokens generated by the encoder (high-level semantics) to the primary semantics of the decoder. Given the significant differences in semantic representation between different layers of large language models (LLMs), the LSA is trained via full fine-tuning.

$$\widetilde{m} = \mathcal{F}_{k_{\text{LSA}}}(\widetilde{H}), \tag{3}$$

where $H$ is the final compressed representation, $\mathcal{F}_{k_{\text{LSA}}}$ denotes Transformer blocks initialized with the weights from the first $k$ layers of the decoder, and $\widetilde{m}$ denotes the generated soft tokens. Just one layer of LSA is sufficient to achieve excellent semantic preservation (for space limitations, please refer to Appendix B), so in this work, we can just set $k_{\text{LSA}} = 1$.

### 3.4 Autoencoder Training

The Autoencoder Training process, which aims to encode the complete information of the original text into memory embeddings, is achieved through autoencoder-based training. We hope to minimize the loss of the reconstructed text, which can be expressed as:

$$\mathcal{L}_{AE} = -\sum_{i=1} \log p_\phi\left(x_i \mid \widetilde{m}, X^{\text{ins}}, x_{<i}\right), \tag{4}$$

where $p_\phi(\cdot)$ is the decoder probability distribution obtained after the softmax function, and $x_i$ is the $i$-th token in the original text.

### 3.5 Knowledge Extraction

#### 3.5.1 Knowledge Extraction Process

Through autoencoder training, we can ensure that the soft tokens obtained via the encoder and LSA contain complete semantic information. Therefore, the next challenge to address is: *how to extract knowledge from the existing soft tokens?*

To guarantee that the generated soft tokens always retain adequate information, we **freeze** the encoder and LSA during the knowledge extraction process, allowing the decoder to complete knowledge extraction (KE).

We only train the decoder's self-attention module. As shown in Figure 3, the $i$-th token decoding progress can be formulated as:

$$\texttt{Decoder}(\underbrace{\tilde{m}_1, \tilde{m}_2, \tilde{m}_3, \tilde{m}_4, ..., \tilde{m}_{k-1}, \tilde{m}_k}_{\text{soft tokens from the encoder}}, \underbrace{q_1, q_2, ..., q_n}_{\text{question tokens}}, \underbrace{a_1, a_2, ..., a_{i-1}}_{\text{answer tokens}}). \tag{5}$$

Let $d$ denote the decoder's hidden size, $H \in \mathbb{R}^{(k+n+i-1) \times d}$ denote input hidden states to the self-attention module of the decoder in an arbitrary layer. The above hidden states will be projected into queries, keys, and values as follows:

$$\boldsymbol{Q} = \boldsymbol{W}_Q H, \quad \boldsymbol{K} = \boldsymbol{W}_K H, \quad \boldsymbol{V} = \boldsymbol{W}_V H, \tag{6}$$

where $\boldsymbol{W}_Q$, $\boldsymbol{W}_K$, and $\boldsymbol{W}_V$ are the projection heads for knowledge extraction. Thus, we now formally present our self-attention computation:

$$\boldsymbol{V}' = \text{softmax}\left(\text{mask}\left(\frac{\boldsymbol{Q}\boldsymbol{K}^T}{\sqrt{d}}\right)\right)\boldsymbol{V}, \tag{7}$$

where $\boldsymbol{V}'$ denotes the output of the self-attention mechanism, which is a refined, context-aware representation of the input values $\boldsymbol{V}$ after applying attention weights.

#### 3.5.2 Knowledge Extraction Fine-tuning

After completing autoencoder training, we need to teach the decoder how to utilize the soft tokens. We achieve this by performing full fine-tuning of the $\boldsymbol{W_Q}$, $\boldsymbol{W_K}$, and $\boldsymbol{W_V}$ projection matrices in each layer of the decoder, which can be specifically expressed as:

$$\mathcal{L}_{\text{KE}} = -\sum_{i=1}^{n} \log p_\phi\left(a_i \mid \widetilde{m}, q_1, q_2, ..., q_n, a_{<i}\right), \tag{8}$$

where $p_\phi(\cdot)$ is the decoder probability distribution obtained after the softmax function, and $a_i$ denotes the $i$-th token in the predicted answer.

## 4 Experiments

In this section, we attempt to answer the following research questions (RQs): (1) How effective is GMSA in context restoration? (RQ1) (2) How does GMSA utilize knowledge compared with other baselines? (RQ2) (3) How effective are the individual components of GMSA? (RQ3)

### 4.1 Settings

**Training.** GMSA involves a two-stage training process: autoencoder training and knowledge extraction fine-tuning (KEFT). We use four datasets: PwC [7], NaturalQuestions [18], 2WikiMQA [8], and HotpotQA [31] (For more details about the dataset, please refer to Appendix D). Among them, we use PwC to evaluate the performance of context restoration, while the three QA datasets are employed to measure downstream knowledge application. We conduct two separate trainings on the PwC dataset and another on a mixed dataset composed of NaturalQuestions, 2WikiMQA, and HotpotQA. During training, we randomly sampled compression rates (i.e., 4x compression and 8x compression) for each training sample. Due to space constraints, detailed training settings can be found in Appendix C.

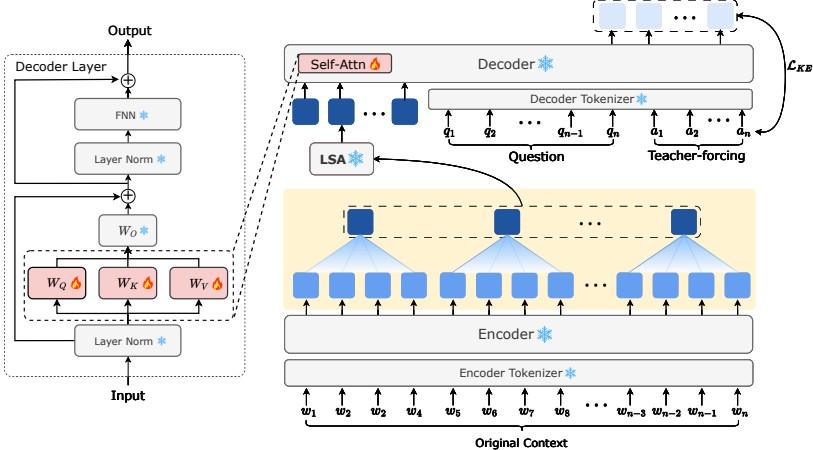

Figure 3: **The process of Knowledge Extraction Fine-tuning (KEFT).** By fine-tuning only the $W_Q$, $W_K$, and $W_V$ in the attention module of the decoder while keeping other modules frozen, the decoder performs teacher-forcing training using soft tokens $\tilde{m}$, question tokens, and the ground truth answer.

**Implementation.** GMSA is implemented based on LLaMA-2-7B (Chat)[2] and Qwen2.5-7B (Instruct). The maximum input length was set to 512 for PwC training and 3072 for NaturalQuestions, 2WikiMQA, and HotpotQA training. To ensure fair comparison, all baseline results are obtained from our re-implementations based on official open-source code.

**Evaluation Metrics.** For the context restoration task on the PwC dataset, we use BLEU [22], Prefix Exact Match, BERT Score [35], and ROUGE [17] for evaluation. For the QA tasks on Natural Questions, TriviaQA, and 2WikiMQA, we use Acc [18], Exact Match (EM) [13], and F1 [31] for evaluation.

**Baselines.** For the task of context restoration, we train a **T**raditional **C**ompression **P**aradigm **A**uto**E**ncoder (i.e., TCP-AE, see Appendix E for details) as a baseline using autoencoder training and the same training hyperparameters as GMSA. In terms of downstream knowledge application, we conduct comprehensive comparisons with various methods in text compression and KV-cache compression fields, including: hard prompt compression (e.g., LongLLMLingua [10], LLMLingua-2-large [21]), soft prompt compression (e.g., AutoCompressor [6], ICAE [7]), and KV-cache compression approaches (e.g., StreamLLM [29], SnapKV [15], Activation Beacon [34]).

## 4.2 Main Result

We highlight the findings of GMSA in two aspects: context restoration and downstream knowledge application.

**For RQ1, in the context restoration task, GMSA-AE significantly outperforms the Traditional Compression Paradigm AutoEncoder (TCP-AE) in multiple aspects, including restoration quality (see Figure 4), convergence speed, and robustness (see Figure 5).** As shown in Figure 4, GMSA-AE outperforms TCP-AE in all evaluation metrics. BLEU Score [22], Prefix Exact Match[3], and ROUGE [17] are token-matching-based metrics, and GMSA-AE's performance in these metrics is at least 20% higher than TCP-AE under all compression constraints, indicating that GMSA-AE has a stronger ability to precisely remember each token. The BERT Score F1 [35], which measures semantic similarity and reflects the model's ability to remember overall semantics, is also about 5% higher for GMSA-AE than TCP-AE. As shown in Figure 5, GMSA-AE converges much faster than

---

[2]We use LLaMA-2 as the default base model unless otherwise specified, because AutoCompressor, ICAE, and Activation Beacon are all based on it, and they are all important baseline models.

[3]Prefix Exact Match represents the ratio of the correctly matched prefix length to the total length. For example, in a 512-token sequence, if the first 128 tokens are an exact match but the 129th token is not, the Prefix Exact Match score is calculated as 128/512 = 0.25.

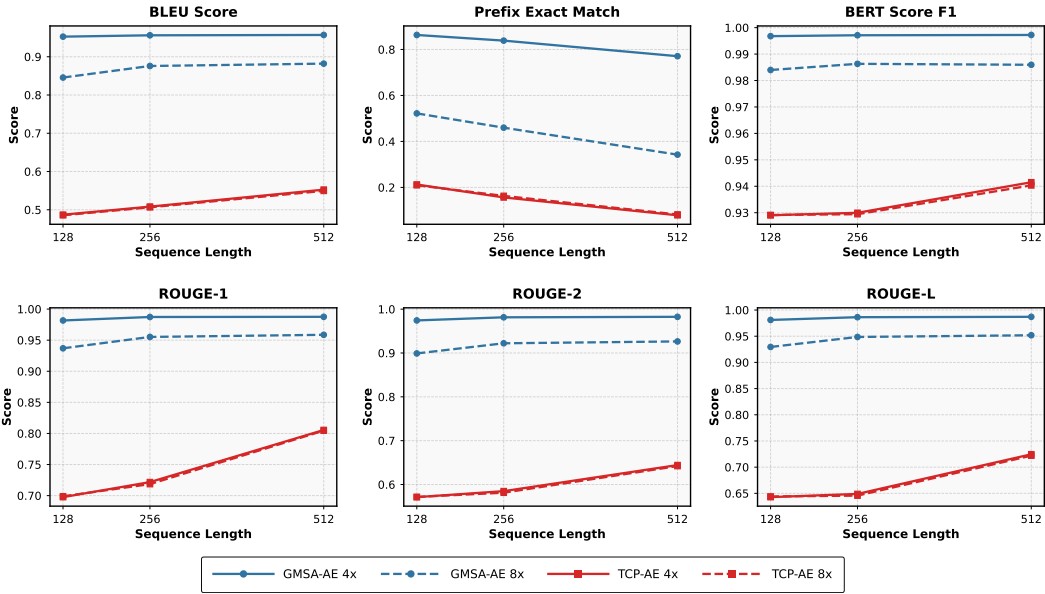

Figure 4: **GMSA-AE v.s. TCP-AE on the context restoration task.** Sequence Length represents different context restoration lengths (i.e., 128, 256, 512), and the models are trained with a maximum length of 512.

TCP-AE. GMSA-AE convergence around 1000 training steps, while TCP-AE has not fully converged even after 5000 steps. Moreover, significantly reducing the number of encoder layers (e.g., to 8 encoder layers) makes TCP-AE converge much more slowly. In contrast, GMSA-AE demonstrates robustness under different settings. In terms of convergence speed, reducing the number of encoder layers even further accelerates the convergence of GMSA-AE: versions with 8 or 16 encoder layers converge faster than those with 32 layers, possibly because the cross-layer semantic alignment challenge is alleviated with fewer encoder layers. From the perspective of semantic retention, the Average BERT Score F1 of different encoder layers remains consistent under various compression rates, indicating that even with a small number of encoder layers (e.g., 8 layers), GMSA-AE can still effectively retain semantic information and complete high-quality memory tasks. We also evaluate the quality of the reconstructed text using perplexity, and the results show that GMSA-AE significantly outperforms TCP-AE (see Appendix I). Moreover, we conduct specific case studies to further verify the performance gap between GMSA-AE and TCP-AE (see Appendix H).

**For RQ2, GMSA demonstrates significantly better performance than other baselines under various compression rate constraints (see Table 1).** In the KV-cache compression methods, the compressed representation and the target model must be consistent. Although this avoids the problem of cross-layer semantic alignment, it severely limits the flexibility of applying the compressed representation. Compared with the KV-cache compression methods (i.e., streamLLM, SnapKV, and Activation Beacon), GMSA achieves the best performance while maintaining flexibility. In contrast to prompt-based compression algorithms, whether they are query-independent prompt compression algorithms (i.e., ICAE, AutoCompressor, and LLMLingua-2-large) or query-dependent LongLLMLingua, their performance is far below that of GMSA. It is worth noting that GMSA adopts a query-independent compression mechanism and still significantly outperforms the query-dependent LongLLMLingua, which sufficiently illustrates the effectiveness and superiority of GMSA.

### 4.3 Efficiency Analysis

In this section, we discuss the efficiency of our proposed method. By using soft tokens instead of the long original context to enhance the inference process, our method reduces the inference cost of the original context during the generation process by a factor of $r$. The overall floating-point operations (FLOPs) are calculated through two processes: compression and generation.

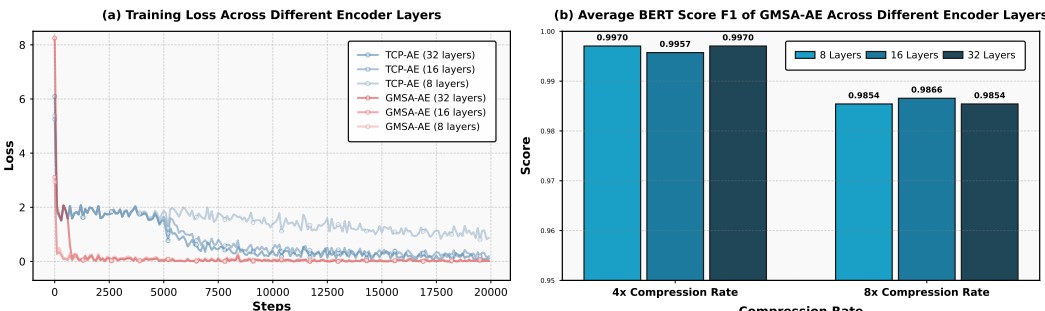

Figure 5: **Analysis of the Effectiveness of Different Encoder Layers.** (a) represents the comparison of convergence speed between GMSA-AE with different encoder layers and TCP-AE. (b) denotes the impact of different encoder layers on the semantic retention of GMSA-AE. The average BERT Score F1 refers to the average F1 score across different context restoration lengths (i.e., 128, 256, and 512).

Table 1: Experimental results on three QA benchmark datasets. We **bold** the optimal and underline the suboptimal of baselines. **Acc** refers to accuracy, **EM** refers to exact match, and **F1** refers to the F1 score. **Closed-book** indicates using only the input question as the input, while **Original Prompt** indicates using all retrieved documents as the input.

| Methods | NaturalQuestions | | | 2WikiMQA | | | HotpotQA | | |
|---|---|---|---|---|---|---|---|---|---|
| | Acc | EM | F1 | Acc | EM | F1 | Acc | EM | F1 |
| Closed-book | 24.14 | 20.23 | 21.88 | 25.37 | 24.96 | 27.82 | 18.34 | 17.22 | 24.02 |
| Original Prompt | 55.40 | 15.07 | 26.81 | 37.54 | 30.84 | 37.79 | 44.21 | 34.35 | 47.49 |
| *4x compression constraint* | | | | | | | | | |
| *KV-cache Compression Methods* | | | | | | | | | |
| StreamLLM [29] | 29.53 | 7.87 | 15.38 | 28.47 | 26.49 | 30.78 | 28.90 | 23.87 | 34.32 |
| SnapKV [15] | 58.64 | 12.58 | 23.07 | 29.86 | 27.61 | 32.62 | 37.35 | 30.51 | 42.08 |
| Activation Baecon [34] | 56.20 | 25.65 | 34.17 | 34.45 | 24.42 | 32.05 | 44.45 | 25.80 | 39.82 |
| *Prompt Compression Methods* | | | | | | | | | |
| AutoCompressor [6] | 13.79 | 0.00 | 1.34 | 41.56 | 0.00 | 8.07 | 20.98 | 0.01 | 6.80 |
| ICAE [7] | 17.33 | 1.24 | 7.05 | 35.17 | 10.25 | 22.04 | 34.16 | 13.02 | 26.69 |
| LongLLMLingua [10] | 53.41 | 39.62 | 43.03 | 33.88 | 31.71 | 37.05 | 40.31 | 35.55 | 48.68 |
| LLMLingua-2-large [21] | 41.77 | 29.49 | 34.79 | 31.07 | 28.88 | 33.37 | 33.15 | 28.80 | 40.89 |
| **GMSA** | **69.98** | **58.12** | **57.59** | **55.95** | **49.55** | **57.17** | **53.52** | **44.60** | **59.31** |
| *8x compression constraint* | | | | | | | | | |
| *KV-cache Compression Methods* | | | | | | | | | |
| StreamLLM [29] | 31.22 | 7.72 | 14.93 | 27.43 | 25.82 | 29.76 | 26.58 | 21.78 | 32.21 |
| SnapKV [15] | 57.21 | 11.86 | 22.49 | 28.19 | 26.56 | 30.97 | 34.54 | 28.10 | 40.16 |
| Activation Baecon [34] | 51.22 | 23.01 | 31.45 | 33.20 | 25.12 | 32.20 | 40.30 | 24.40 | 37.63 |
| *Prompt Compression Methods* | | | | | | | | | |
| AutoCompressor [6] | 17.51 | 0.00 | 1.63 | 41.76 | 0.00 | 8.09 | 22.04 | 0.00 | 6.93 |
| ICAE [7] | 17.74 | 0.72 | 3.23 | 33.56 | 5.74 | 17.19 | 30.40 | 4.42 | 15.80 |
| LongLLMLingua [10] | 46.55 | 36.65 | 40.72 | 31.53 | 29.93 | 34.08 | 34.73 | 31.60 | 43.85 |
| LLMLingua-2-large [21] | 30.73 | 21.92 | 27.61 | 27.45 | 26.57 | 29.64 | 24.14 | 22.11 | 31.69 |
| **GMSA** | **62.34** | **51.00** | **53.09** | **51.33** | **46.67** | **54.22** | **46.52** | **38.39** | **53.77** |

The compression process can be expressed as:

$$\text{FLOPs}^{comp} = F^{\text{Encoder}}(L) + F^{\text{LSA}}\left(\left\lceil \frac{L}{r} \right\rceil\right)$$

Here, $L$ denotes the original context length, $L_q$ denotes the question length, and $F^*(\cdot)$ represents the FLOPs complexity measure for module $*$, with the specific calculation process detailed in Appendix F. The symbol $*$ indicates the architectural components, where $* \in \{\text{Decoder}, \text{Encoder}, \text{LSA}\}$. For the generation process, assuming the answer length is $L_a$, the generation process requires $L_a$ forward passes. The FLOPs for the $i$-th forward pass are given by:

$$\text{FLOPs}_i^{forward} = F^{\text{Decoder}} \left( \left\lceil \frac{L}{r} \right\rceil, L_q, i \right)$$

Combining the costs of all components, the total FLOPs complexity is:

$$\text{FLOPs} = \sum_{i=1}^{L_a} \text{FLOPs}_i^{\text{forward}} + \text{FLOPs}^{comp}$$

Thanks to the ability to retain complete semantics with only a few encoder layers (e.g., 8 layers), **GMSA achieves the lowest end-to-end inference latency, which is approximately 2x faster than other methods (see Appendix G).**

### 4.4 Ablation Study

**For RQ3, to investigate the impact of each component in GMSA, we conduct the following four ablation experiments (see Table 2):** (1) Ours w/o Autoencoder Training refers to performing knowledge extraction fine-tuning on GMSA directly without knowledge memory training. (2) Ours w/o Knowledge Extraction Fine-tuning means only performing Autoencoder-Training on GMSA. (3) Ours w/o Group Merging indicates that we replace group merging with appending meaningless learnable tokens when generating summary vectors. (4) Ours w/o Layer Semantic Alignment means we do not use the

Table 2: The impact of different components in GMSA on the PwC test set under 4x compression constraint, measured by BERT Score F1.

| Method | BERT Score F1 |
|---|---|
| Default | 0.91 |
| w/o Autoencoder Training | 0.87 |
| w/o Knowledge Extraction Fine-tuning | 0.83 |
| w/o Group Merging | 0.82 |
| w/o Layer Secmantic Alignment | 0.84 |
| w Qwen2.5-7B-Instruct | 0.91 |

Layer Semantic Alignment module and directly employ summary vectors as soft tokens. (5) Ours w/ Qwen2.5-7B-Instruct refers to replacing the decoder with Qwen2.5-7B-Instruct.

**In summary, the removal of any single component leads to a significant drop in performance, which fully demonstrates the necessity and effectiveness of each component.** Removing Autoencoder Training makes it difficult for GMSA to generate summary vectors that encompass complete semantics, while eliminating Knowledge Extraction Fine-tuning causes GMSA to lose its ability to extract knowledge in downstream tasks, both of which would deteriorate performance. Replacing Group Merging with appending learnable tokens would increase the difficulty of learning, and discarding the Layer Semantic Alignment module would result in misalignment between the high-level semantic information represented by summary vectors and the low-level semantic space of the decoder's input. When the encoder and decoder are different, GMSA can still maintain high performance, which fully demonstrates its robustness and generalization ability.

## 5 Conclusion

This paper introduces GMSA, a context compression framework based on an encoder-decoder structure. It effectively and efficiently learns summary vectors and bridges the significant gap between the semantics representation of different layers via group merging, and a layer semantic alignment (LSA) module. GMSA first undergoes autoencoder training to ensure that the generated soft tokens contain complete semantics, and then adapts to downstream tasks through knowledge extraction fine-tuning (KEFT). Experiments demonstrate that GMSA converges quickly, can stably converge even with random sampling compression rates for each sample and using only a few encoder layers, and has excellent context restoration capabilities. It outperforms existing baselines by a large margin in downstream tasks, paving the way for the efficient application of LLMs.

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

# A  Related Works

**Hard Prompt Compression.**  Hard prompt compression refers to the removal of some less important tokens from the original prompt or the generation of summaries to achieve compression. The compressed prompt is explicit text. It can mainly be divided into the following four categories: (1) Perplexity-based methods. Selective-Context [14] removes certain lexical units based on perplexity, while methods such as LLMLingua [11], LongLLMLingua [10], and Perception Compressor [25] adopt a coarse-to-fine framework to gradually eliminate less important parts. (2) Bidirectional semantic-based methods. Considering the unidirectional nature of perplexity, some approaches employ bidirectional semantic information for compression, such as LLMLingua-2 [21], MOOSComp [38], and EFPC [3]. (3) Methods based on intrinsic attention mechanisms. Compression is achieved through the intrinsic attention mechanisms of LLMs, such as PIS [4] and AttnComp [37]. (4) Summary generation. This involves generating linguistic summaries that contain useful information for long text content, such as CompACT [33] and RECOMP [30]. Although these methods improve the computational efficiency of inference through prompt compression, they compromise the semantic integrity of the original prompt.

**Soft Prompt Compression.**  Soft prompt compression has become a research hotspot in the field of Natural Language Processing (NLP). The goal of soft prompt compression is to learn a set of soft tokens (with a sequence length much shorter than the original text) to achieve compression, where the compressed soft prompts cannot be explicitly converted into text. Among existing methods, xRAG [5] focuses on processing short texts and extreme compression. More mainstream methods, such as GIST [20], AutoCompressor [6], 500xCompressor [16], ICAE [7] and VoCo-LLaMA [32], learn soft tokens in an autoregressive manner by appending randomly initialized additional tokens. This leads to the semantics of anchor tokens in the input sequence being increasingly emphasized layer by layer, while the semantics of other tokens are diluted and cannot be fully preserved in the summary vectors. Moreover, these methods only use Multilayer Perceptrons (MLPs) for coarse-grained semantic alignment when semantic alignment is required, ignoring the significant differences in representations across different layers of large models. *Our proposed method efficiently and effectively extracts summary vectors through group merging. By employing a group average pooling merging strategy, it addresses the issue of uneven semantic retention. Additionally, it bridges the semantic gap between different layers of large models through a Layer Semantic Alignment (LSA) module.*

**KV-cache Compression.**  Research in this direction focuses on directly compressing the KV-cache in each transformer layer, considering factors such as layer-wise compression, attention heads, the importance of different KVs, or token-level approaches. Examples include CLA [2], which shares KV-cache across layers; GQA [1] and MQA [24], which reduce the number of heads for keys and values; StreamLLM [29] and SnapKV [15], which discard unimportant KVs for efficient compression; and Activation Beacon, which appends some meaningless tokens (shorter than the original length) and learns compressed representations in the KV-cache of these tokens for each layer. While KV-cache-based compression methods can accelerate inference, they require the compression and response models to be identical. This limitation restricts practical applications and increases resource consumption—for instance, in prompt compression for large models (e.g., 70B), a smaller model (e.g., 7B) cannot be used as the compression model; instead, the same oversized model must be employed.

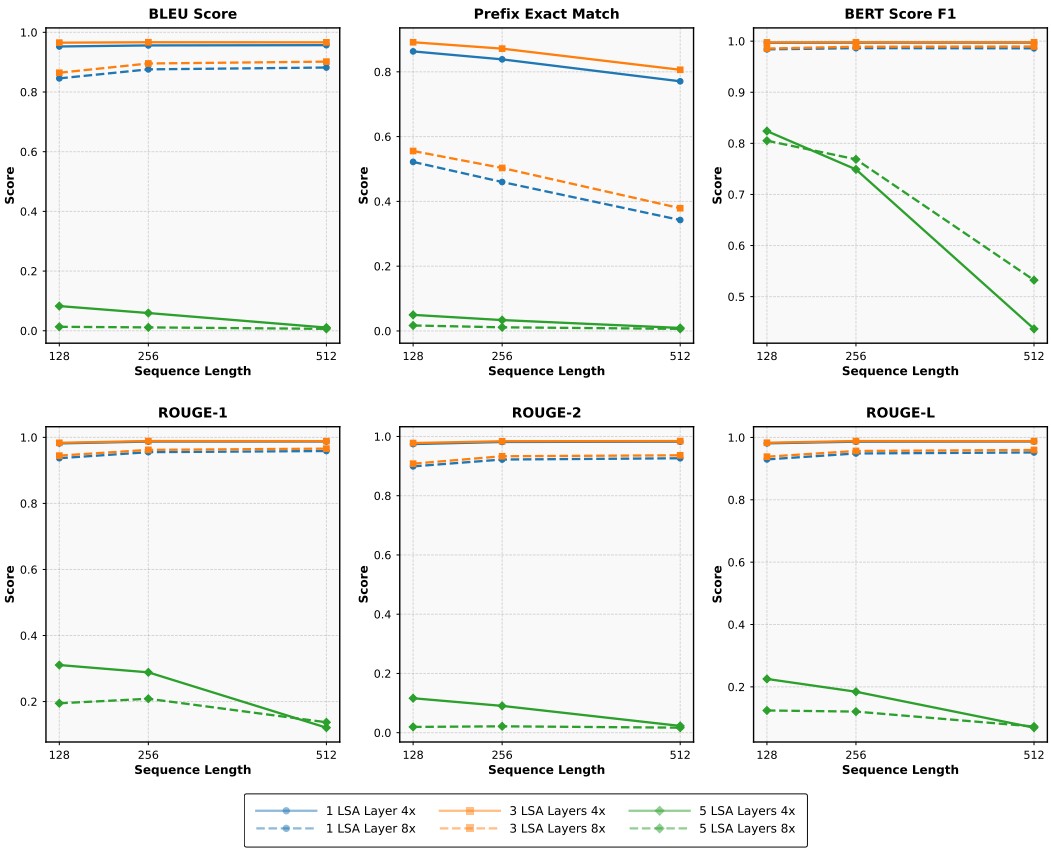

Figure 6: **The impact of different layers of LSA on semantic retention in GMSA-AE.** Sequence Length represents different context restoration lengths (i.e., 128, 256, 512), and the model is trained with a maximum length of 512.

## B Impact of different Layer Semantic Alignment layers

We conduct experiments to investigate the impact of layer semantic alignment (LSA) module with varying numbers of layers on the retention of complete semantics, and the results are shown in Figure 6. We can draw the following conclusions: (1) Only one layer of LSA is sufficient to achieve good retention of complete semantics (with a BERT Score F1 close to 1, and it already performs the best among different numbers of LSA layers). (2) When the number of LSA layers becomes too high, e.g., using five layers of LSA, it may actually lead to a decrease in the GMSA's ability to retain semantics. This is likely because as the LSA module becomes deeper, it contains more high-layer semantics and fewer low-layer semantics, thereby increasing the difficulty of semantic alignment.

## C Implementation Details

We train GMSA on two NVIDIA A100 GPUs (80GB) using bf16 precision. For the PwC dataset, we train on the full dataset with 10,000 steps for Autoencoder Training and 5,000 steps for Knowledge Extraction Fine-tuning (KEFT). For the QA datasets (i.e., NaturalQuestions, 2WikiMQA, and HotpotQA), we sample 15,000 examples from each to form the training set, using 20,000 steps for Autoencoder Training and 1,000 steps for KEFT, respectively. Other parameters are listed in Table C.

Table 3: Training Hyperparameters.

| Hyperparameter | Value |
|---|---|
| Optimizer | AdamW |
| Learning Rate | $1 \times 10^{-4}$ (Autoencoder Training) |
| | $1 \times 10^{-5}$ (KEFT) |
| Batch Size | 4 (Autoencoder Training) |
| | 16 (KEFT) |
| Scheduler | Linear |
| Gradient Clip Norm | 2.0 |

## D  Datasets Details

**PwC dataset.**  In the PwC dataset [7], each sample is a triplet (context, prompt, answer), where the context is sampled from the Pile and the prompt and answer are generated by GPT-4. The training set contains 241,564 samples, the test set contains 18,146 samples, and the average token length of the dataset is 609[4].

**NaturalQuestions.**  NaturalQuestions [18], in which each question corresponds to 20 relevant documents, 19 of which are distractors and only one contains the ground truth answer. The training set contains 75,322 samples, the test set contains 2,655 samples, and the average token length of the dataset is 3,253.

**HotpotQA.**  HotpotQA [31] is a two-hop reasoning dataset, where the answers are scattered across two documents. Specifically, each question corresponds to 10 relevant documents, two of which are the ground truth documents. The training set contains 89,609 samples, the test set contains 7,345 samples, and the average token length of the dataset is 1,567.

**2WikiMQA.**  Compared with HotpotQA, 2WikiMQA [8] includes more complex reasoning paths, and the combination of structured and unstructured data, usually involving two or more hops and having higher difficulty. The training set contains 167,454 samples, the test set contains 12,576 samples, and the average token length of the dataset is 1098.

## E  Traditional Compression Paradigm Autoencoder Training

As shown in Figure 7, to fully measure the context restoration capability of GMSA after Autoencoder Training, we conduct Autoencoder Training following the traditional compression paradigm, using the same training method as GMSA (i.e., randomly sampling compression rates for training examples and other hyperparameters in the training process are also the same) to obtain Traditional Compression Paradigm Autoencoder (TCP-AE)[5].

## F  FLOPs Calculation

Let $L_{\text{in}}$ denote the input sequence length. We calculate the floating-point operations (FLOPs) for a single layer can be decomposed into Attention and Feed Forward Network (FFN) operations. The calculation process for the Attention operation is:

---

[4]We uniformly use the tokenizer of LLaMA-2-7B (chat) to calculate the token length of the text.

[5]The entire structure is similar to the pretraining structure of ICAE, but the training paradigm is different. For example, we randomly sample the compression rate for training, which increases the difficulty of training.

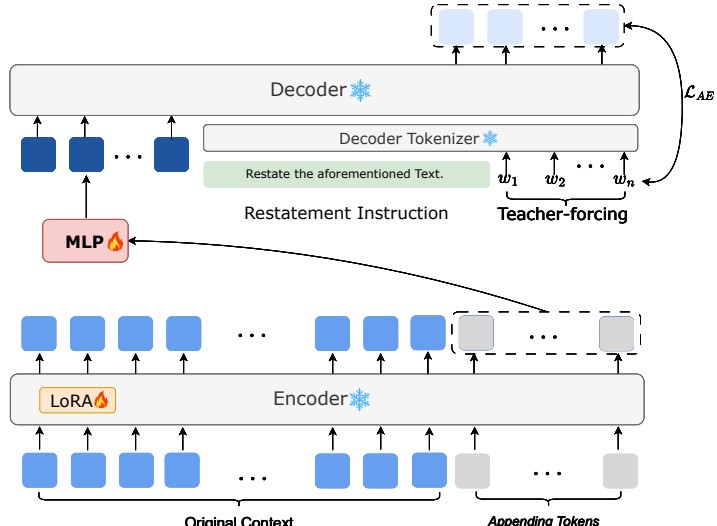

Figure 7: **The training process of Traditional Compression Paradigm Autoencoder (TCP-AE).** The traditional compression paradigm first adds appending tokens after the Original Context, then employs an encoder (e.g., LLaMA) to autoregressively learn summary vectors. These summary vectors are then processed through a Multilayer Perceptron (MLP) layer to achieve coarse-grained semantic alignment, resulting in soft tokens. On the decoder side, context restoration training is conditioned on soft tokens, with cross-entropy used as the final loss.

$$
\begin{aligned}
F^{Attention}(L_{\text{in}}) &= F^{qkv}(L_{\text{in}}) + F^{qk}(L_{\text{in}}) + F^{softmax}(L_{\text{in}}) + F^{av}(L_{\text{in}}) + F^{out}(L_{\text{in}}) \\
F^{qkv}(L_{\text{in}}) &= 2 \times L_{\text{in}} \times D \times d \times h^q + 2 \times 2 \times L_{\text{in}} \times D \times d \times h^k \\
F^{qk}(L_{\text{in}}) &= 2 \times h^q \times L_{\text{in}} \times L_{\text{in}} \times d \\
F^{softmax}(L_{\text{in}}) &= h^q \times L_{\text{in}} \times L_{\text{in}} \\
F^{av}(L_{\text{in}}) &= 2 \times h^q \times L_{\text{in}} \times L_{\text{in}} \times d \\
F^{out}(L_{\text{in}}) &= 2 \times L_{\text{in}} \times d \times h^q \times D
\end{aligned}
\tag{9}
$$

The calculation process for the FFN can be formulated as:

$$
\begin{aligned}
F^{FFN}(L_{\text{in}}) &= F^{up}(L_{\text{in}}) + F^{down}(L_{\text{in}}) \\
F^{up}(L_{\text{in}}) &= 2 \times L_{\text{in}} \times D \times 2 \times I \\
F^{down}(L_{\text{in}}) &= 2 \times L_{\text{in}} \times D \times I
\end{aligned}
\tag{10}
$$

Denote the original context length as $L$, the compression rate as $r$, question length as $L_q$, answer length as $L_a$, the number of layers in the LSA as $N_{\text{LSA}}$, the number of decoder layers as $N_{\text{Dec}}$, number of encoder layers as $N_{\text{Enc}}$, query head number as $h^q$, key/value head number as $h^k$, the hidden size as $D$, head dimension as $d$, intermediate size as $I$, and vocabulary size as $V$. Therefore, the FLOPs of the encoder, LSA, and decoder can be expressed as:

$$
\begin{aligned}
F^{Encoder}(L) &= \left( F^{Attention}(L) + F_E^{FFN}(L) \right) \times N_{\text{Enc}} \\
F^{LSA}(\lceil L/r \rceil) &= \left( F_L^{Attention}(\lceil L/r \rceil) + F_L^{FFN}(\lceil L/r \rceil) \right) \times N_{\text{LSA}} \\
F^{Decoder}(\lceil L/r \rceil, L_q, L_a) &= \sum_{i=1}^{L_a} \left( F_D^{Attention}(\lceil L/r \rceil, L_q, i) + F_D^{FFN}(\lceil L/r \rceil, L_q, i) \right) \times N_{\text{Dec}}
\end{aligned}
\tag{11}
$$

Table 4: **Latency Evaluation.** Latency evaluation of different methods under varying compression constraints on the Natural Questions dataset. The symbol ✗ indicates that the specific processing time is unavailable.

| Method | Compression Time | Decoding Time | End-to-End Inference Time |
|---|---|---|---|
| Original Context | - | 1.14 | 1.14 |
| *4x compression constraint* | | | |
| StreamLLM | ✗ | ✗ | 1.47 |
| SnapKV | ✗ | ✗ | 0.99 |
| Activation Beacon | ✗ | ✗ | 3.06 |
| ICAE | 0.73 | 1.06 | 1.79 |
| **GMSA** | **0.27** | **0.18** | **0.45** |
| *8x compression constraint* | | | |
| StreamLLM | ✗ | ✗ | 1.41 |
| SnapKV | ✗ | ✗ | 0.99 |
| Activation Beacon | ✗ | ✗ | 1.92 |
| ICAE | 0.56 | 2.60 | 3.16 |
| **GMSA** | **0.27** | **0.15** | **0.42** |

where $N_{\text{Enc}} \ll N_{\text{total}}$ uses only shallow layers (e.g., 8/32 in LLaMA), $N_{\text{LSA}}$ is generally set to 1 follows from LSA's layer-agnostic property (see Appendix B), and $r > 1$ represents standard compression rates.

# G   Latency Evaluation

We conduct an empirical test on the Natural Questions dataset to evaluate the impact of GMSA on inference efficiency under 4x and 8x compression constraints.[6] In this efficiency test, we fix the generation length to 100. Table 4 shows that the context compression by GMSA helps improve the inference efficiency of LLMs. Compared with all settings, including the original prompt, Kv-cache compression algorithms (i.e., StreamLLM, SnapKV, and Activation Beacon), and the encoder-decoder-based ICAE, GMSA achieves more than a 2x end-to-end inference speedup.

# H   Perplexity Evaluation

For the task of context restoration, we evaluate model performance from the perspective of perplexity. The experimental results are shown in Table 5. Based on our analysis, we have two key findings: (1) Under different compression constraints and restoration lengths, the perplexity of the recovered text conditioned on TCP-AE-generated soft tokens is significantly higher than that of the recovered text conditioned on the Original Context. (2) Except for the case where the compression constraint is 8x and the restoration length is 512, where GMSA-AE's recovered text perplexity is slightly lower than that of the Original Context (by only 0.02), in all other cases, GMSA-AE's recovered text perplexity is lower than that of the Original Context. Furthermore, in all scenarios, GMSA-AE's recovered text perplexity is significantly lower than that of the recovered text conditioned on TCP-AE-generated soft tokens.

# I   Case Study

As shown in Table 6, we use the restoration of a specific text to study the performance of GMSA-AE in context restoration. In the restored text, GMSA-AE only has the last word inconsistent with the

---

[6]We test the latency on two NVIDIA A800 GPUs (80G).

Table 5: **Comparison of the average token perplexity under different condition types on the PwC test set.** "Condition Type" represents the basic conditions under which the large language models (LLMs) recovers the text, which are divided into three types: recovering from the Original Context, recovering from the soft tokens generated by TCP-AE, and recovering from the soft tokens generated by GMSA-AE. Different Sequence Lengths represent different lengths of the context restoration task.

| Condition Type | Sequence Length | | |
|---|---|---|---|
| | 128 | 256 | 512 |
| Original Context | 1.12 | 1.06 | 1.03 |
| *4x compression constraint* | | | |
| TCP-AE | 1.36 | 1.34 | 1.35 |
| **GMSA-AE** | **1.01** | **1.01** | **1.00** |
| *8x compression constraint* | | | |
| TCP-AE | 1.36 | 1.34 | 1.35 |
| **GMSA-AE** | **1.08** | **1.06** | **1.05** |

original text, i.e., restoring "it" to its plural form "they". In contrast, TCP-AE not only exhibits inconsistencies in some word expressions (such as "medication" and "drugs") but also displays large segments of discrepancies with the original text.

# J   Limitations

Although GMSA demonstrates strong performance and achieves significant inference acceleration, it requires two-stage training, i.e., autoencoder training and knowledge extraction fine-tuning (KEFT), to adapt to downstream tasks. Therefore, GMSA has certain requirements for GPU resources. Due to limited computational resources, i.e., two NVIDIA A800 80G GPUs, GMSA is evaluated on sequences shorter than 5K in length. In future work, assuming access to more computational resources, we plan to evaluate GMSA on longer sequences.

Table 6: **An example showing GMSA-AE and TCP-AE's context restoration performance.** Text highlighted in ==yellow== indicates discrepancies from the **Original Context**.

| Origin Context | GMSA-AE | TCP-AE |
|---|---|---|
| Craig F. Walker \| Boston Globe \| Getty Images
The Trump administration is making good on its latest effort to lower out-of-pocket drug costs for Medicare recipients, but its approach could also limit drug options or even risk eliminating coverage of some prescriptions. The Centers for Medicare and Medicaid Services proposed late Monday changes to Medicare Advantage and Medicare Part D. Among the changes, it would allow insurers to stop covering certain "protected" classes of drugs used to treat common ailments like depression, cancer and HIV. When Congress added a prescription drug benefit to Medicare in 2003, it required insurers to cover at least two different drugs to treat any particular ailment. It also set aside six protected classes of medication where insurers were required to cover "all or substantially all" of the drugs offered to ensure seniors got whatever treatment they needed, like for cancer. The Trump administration thinks that gives drug manufacturers greater negotiating power on prices. The proposal is meant to give insurers more leverage and drive prices down by allowing them to drop coverage of certain drugs in a protected class if it | Craig F. Walker \| Boston Globe \| Getty Images
The Trump administration is making good on its latest effort to lower out-of-pocket drug costs for Medicare recipients, but its approach could also limit drug options or even risk eliminating coverage of some prescriptions. The Centers for Medicare and Medicaid Services proposed late Monday changes to Medicare Advantage and Medicare Part D. Among the changes, it would allow insurers to stop covering certain "protected" classes of drugs used to treat common ailments like depression, cancer and HIV. When Congress added a prescription drug benefit to Medicare in 2003, it required insurers to cover at least two different drugs to treat any particular ailment. It also set aside six protected classes of medication where insurers were required to cover "all or substantially all" of the drugs offered to ensure seniors got whatever treatment they needed, like for cancer. The Trump administration thinks that gives drug manufacturers greater negotiating power on prices. The proposal is meant to give insurers more leverage and drive prices down by allowing them to drop coverage of certain drugs in a protected class if ==they== | Craig Walker \| Boston Globe ==\|== Getty Images
The Trump administration is making good on its latest effort to lower out-of-pocket ==medication== costs for Medicare recipients, but its approach could also limit drug options or even risk eliminating coverage of some prescriptions. The Centers for Medicare and Medicaid Services proposed late Monday changes to Medicare Advantage and Medicare Part D. Among the changes, it would allow insurers to stop covering certain "protected" ==drugs== used to treat common ailments like depression, cancer and HIV. ==The Centers for Medicare and Medicaid Services proposed changes to Medicare Advantage and Medicare Part D. Among the changes, it would allow insurers to stop covering certain drugs that are used to treat common ailments like depression, cancer and HIV. The proposal would have added a prescription drug benefit to Medicare Part B, which currently covers only doctor visits and lab tests. Congress added the prescription drug benefit in 2003 to require insurers to cover at least two different drugs to treat any of the "essential drugs" offered to seniors, regardless of whether they were covered by Medicare== |

