# OpenReview forum: "GMSA: Enhancing Context Compression via Group Merging and Layer Semantic Alignment"
_NeurIPS.cc/2025/Conference — Submitted to NeurIPS 2025_

### Official Review · Reviewer_7eDi · 2025-06-30

**Clarity:** 2
**Significance:** 3
**Originality:** 3
**Rating:** 4
**Confidence:** 3

**Summary:**

This paper introduces GMSA (Group Merge and Semantic Alignment), a context compression framework for large language models (LLMs) that aims to address the inherent computational inefficiency and redundant information problems when dealing with long contexts. The framework adopts an encoder-decoder architecture and contains two key innovations: (1) Group Merge, which efficiently extracts summary vectors by grouping the last hidden states of the encoder and performing average pooling within each group, ensuring unified semantic learning and parallelization; (2) Layer Semantic Alignment (LSA), which bridges the semantic gap between high-level summary representations and low-level decoder inputs via Transformer blocks initialized with low-level decoder weights. The training method consists of two stages: autoencoder training to ensure full semantic preservation in the generated soft tokens, followed by knowledge extraction fine-tuning (KEFT), freezing the encoder and LSA while only fine-tuning the decoder's self-attention module to extract task-specific knowledge.

**Questions:**

Please see the weaknesses.

**Ethical Concerns:**

["NO or VERY MINOR ethics concerns only"]

**Final Justification:**

The authors' response has addressed my concerns, including inadequate explanation, incomplete experimental validation, and lack of flexibility in method design.

**Limitations:**

Yes.

**Quality:**

3

**Strengths And Weaknesses:**

Strengths:

1. Motivations: The paper identifies two core challenges of LLMs in long text processing: the quadratic complexity problem of computational efficiency and the performance degradation caused by redundant information. This problem positioning has practical application value and provides motivation for research.

2. Methods: The Group Merging mechanism effectively avoids the problem of uneven semantic learning in traditional methods by equally considering all tokens in each group. This design not only ensures the integrity of semantics, but also supports parallel processing. The Layer Semantic Alignment (LSA) module alleviates the semantic gap problem between different layers by using the Transformer block initialized with the weights of the low-level decoder.

3. Experiments: The experiments covered two dimensions: context restoration and downstream QA tasks. Multiple datasets (PwC, NaturalQuestions, 2WikiMQA, HotpotQA) were used and compared with multiple baseline methods. Ablation experiments verify the necessity of each component, including autoencoder training, KEFT, Group Merging, and LSA.

4. The paper is easy to follow.

Weaknesses:

1. Although Group Merging and LSA are proposed, there is a lack of in-depth analysis to explain why this design can maintain semantic integrity and how to determine the optimal group size?

2. The experiments were mainly conducted on the 7B model and lacked verification on larger-scale models (such as 13B and 70B), which limits the demonstration of the universality of the method.

3. The experiments mainly focus on QA tasks and lack verification on other NLP tasks (such as summary generation, dialogue systems, code generation, etc.), which limits the versatility of the method.

4. Group Merging essentially destroys the contextual continuity of natural language by forcing continuous token sequences to be divided into groups of fixed size. For example, a complete sentence may be segmented into different groups.

5. The paper claims that "equally considers each group" is an advantage, but this ignores the fact that in real text, the importance of information in different positions is different. Key information and redundant information are treated equally, and this treatment may actually harm the retention of key information.

6. A fixed group size treats all texts equally, but different types of text (technical documents, dialogues, narrative texts, etc.) may require different compression strategies. The paper does not explore how to adaptively adjust the grouping strategy based on content characteristics.

7. The experiments in the paper mainly focus on the final QA performance, but no special experiments are designed to verify whether Group Merging really retains the "complete semantics". For example, there is a lack of semantic similarity analysis and information theory analysis of the compressed representation.

---

> ### Author Rebuttal · Authors · 2025-07-31
>
> # Response to Reviewer 7eDi
>
> **We sincerely thank you for your valuable and insightful comments, which have greatly helped us improve our manuscript. Below, we provide detailed responses to each issue you raised.**
>
> > W1. Although Group Merging and LSA are proposed, there is a lack of in-depth analysis to explain why this design can maintain semantic integrity and how to determine the optimal group size?
> >
>
> Thank you for pointing this out. We clarify and further explain this design as follows:
>
> - **Semantic Integrity:** Group Merging maintains semantic integrity primarily due to uniform token aggregation within groups. Unlike traditional methods that unevenly emphasize certain "anchor" tokens, our method equally aggregates all tokens, effectively preserving diverse semantic nuances. Layer Semantic Alignment (LSA) module further ensures semantic integrity by aligning higher-layer semantic vectors to lower-layer semantic spaces using Transformer blocks initialized with decoder parameters, thus bridging the inherent semantic gaps.
> - **Optimal Group Size:** We selected the group size based on practical compression requirements and empirically validated through ablation studies (Appendix B and Table 2). Our experiments revealed stable performance across multiple group sizes (4x, 8x compression). We recognize your valuable suggestion and will add adaptive group-size optimization based on text characteristics to our future work.
>
> > W2. The experiments were mainly conducted on the 7B model and lacked verification on larger-scale models (such as 13B and 70B), which limits the demonstration of the universality of the method.
> >
>
> Thanks for highlighting this point. We have now conducted additional experiments using LLaMA-2-13B. For simplicity, the performance of method under Nx compression rate constraints will be denoted as 'Nx method' in the following.
>
> **NQ HQA MQA (LLaMA-2-13B as backbone)**
>
> | Methods | NQ | HQA | MQA |
> | --- | :---: | :---: | :---: |
> | **4x GMSA** | **75.4** | **61.8** | **65.6** |
> | 8x GMSA | 68.8 | 54.8 | 60.5 |
> | Original Prompt | 60.8 | 48.3 | 41.1 |
>
> We present these results clearly in the revised version, demonstrating that GMSA's performance gains persist across larger-scale models, indicating strong universality. We plan to conduct further experiments with 70B models when more resources are available.
>
> > W3. The experiments mainly focus on QA tasks and lack verification ... which limits the versatility of the method.
> >
>
> We acknowledge this limitation. To directly address your suggestion, we have conducted additional experiments to evaluate GMSA on summarization (CNN / DailyMail [1]), code completion (Repobench [2]), math capability (GSM8K [3]) a wide range of disciplines (MMLU [4]). We compare GMSA with strong baseline, i.e., Activation Beacon [5].
>
> We trained and evaluated GMSA on the training and test sets of the CNN / DailyMail dataset, respectively. For the Repobench dataset, we split it into training and test sets at a ratio of 4:1 and conducted training and evaluation accordingly.
>
> **CNN / DailyMail (LLaMA-2-7B as backbone)**
>
> | Methods | BERT Score F1 |
> | --- | :---: |
> | 4x Activation Beacon | 87.0 |
> | **4x GMSA** | 89.1 |
> | 8x Activation Beacon | 86.5 |
> | **8x GMSA** | 88.8 |
>
> **Repobench (Qwen-2-7B as backbone)**
>
> | Methods | Code Sim. Score |
> | --- | :---: |
> | 4x Activation Beacon | 23.1 |
> | **4x GMSA** | **34.5** |
> | 8x Activation Beacon | 22.9 |
> | **8x GMSA** | **31.5** |
>
> We directly evaluate the performance of the GMSA version trained on the mixed NQ, HQA, and MQA dataset on MMLU and GSM8K to assess its short-context capability and generalization.
>
> **MMLU**
>
> | Methods | LLaMA-2-7B | Qwen-2-7B |
> | --- | :---: | :---: |
> | Activation Beacon | 45.1 | 64.3 |
> | **GMSA** | **47.6** | **67.3** |
> | Original Prompt | 46.3 | 65.6 |
>
> **GSM8K**
>
> | Methods | LLaMA-2-7B | Qwen-2-7B |
> | --- | :---: | :---: |
> | **Activation Beacon** | **27.8** | **81.9** |
> | GMSA | 26.4 | 80.5 |
> | Original Prompt | 27.6 | 81.3 |
>
> GMSA even outperforms the untrained LLaMA-2-7B on MMLU, demonstrating the superior short-context capability and generalization of our method. However, its performance slightly drops on GSM8K, likely because GMSA was not trained on mathematical data and thus shows somewhat reduced performance in this specific domain.
>
> These new experiments demonstrate that GMSA maintains robust performance across broader generation-centric scenarios.
>
> > W4. Group Merging essentially destroys the contextual continuity of natural language ... may be segmented into different groups.
> >
>
> We recognize the reviewer's concern about potential disruption of continuity. While Group Merging does unavoidably segment natural-language sentences, the generated soft tokens remains continuous in the dense vector space because the relative order of every compressed group mirrors that of the original text. Empirical evidence (Table 2 and the case study in Appendix H) confirms that semantic continuity and integrity are preserved at a high level after merging.
>
> > W5. The paper claims that "equally considers each group" is an advantage ... treatment may actually harm the retention of key information.
> >
>
> Thank you for the insightful question. As Figure 1 illustrates, conventional compression paradigms suffer from uneven semantic learning. GMSA adapts to downstream tasks through a two-stage training process. **In the first stage (i.e., Autoencoder Training) the sole objective is to preserve the complete semantics, so treating every group equally is advantageous;** this uniformity enables the learning of soft tokens that faithfully encode the full semantic content. The second stage (i.e., Knowledge Extraction Fine-Tuning (KEFT))then focuses on teaching the decoder how to extract and utilize the question-relevant key information contained within those soft tokens.
>
> Thank you for the insightful question. As shown in Figure 1, traditional compression paradigm suffers from uneven semantic learning. GMSA addresses this through a two-stage training procedure. **In the first stage (Autoencoder Training), the sole objective is to preserve the semantics of every original token as completely as possible via soft tokens. Hence, treating each group equally is advantageous; this uniformity enables the model to learn soft tokens that carry the full semantic content, rather than over-emphasizing certain key information (which would dilute the semantics of other tokens and hinder complete preservation).** The second stage (Knowledge Extraction Fine-Tuning, KEFT) then focuses on teaching the decoder to extract and leverage the question-relevant key information embedded in those soft tokens.
>
> > W6. A fixed group size treats all texts equally, but different types of text ... how to adaptively adjust the grouping strategy based on content characteristics.
> >
>
> We appreciate the reviewer highlighting this important perspective. In this work, we employed a fixed group size to maintain architectural simplicity and computational efficiency, and our experiments demonstrate that this approach works well across a variety of datasets and domains (e.g., QA on Wikipedia-based passages, multi-hop reasoning, etc.). That said, we fully agree that different text genres may benefit from dynamic or adaptive grouping strategies.
>
> As part of our future work, we plan to explore **content-aware adaptive grouping**, possibly by using shallow classifiers or linguistic heuristics to estimate information density or discourse structure. These adaptive mechanisms could guide more semantically-aligned grouping decisions based on textual features, potentially improving both compression fidelity and downstream performance. We thank the reviewer for encouraging this promising direction.
>
> > W7. The experiments in the paper mainly focus on the final QA performance, but no special experiments are designed to verify whether Group Merging really retains the "complete semantics". For example, there is a lack of semantic similarity analysis and information theory analysis of the compressed representation.
> >
>
> We thank the reviewer for emphasizing the need for deeper validation of semantic retention. In fact, our experiments already include dedicated evaluations of completeness (Figure 4 and Appendix H):
>
> - **Text reconstruction test (Figure 4).** We reconstruct the original context from the soft tokens generated by GMSA-AE and compute both token-matching metrics (e.g., BLEU, ROUGE and Prefix Exact Match) and semantic-matching metrics (e.g., BERT Score F1). The results show high performance on both metrics, and GMSA consistently outperforms the Traditional Compression Paradigm AutoEncoder (TCP-AE) by at least 20 % across different sequence lengths and compression rates.
> - **Perplexity-based information analysis.** We measure the perplexity of the reconstructed text from the soft tokens. GMSA-AE yields consistently lower perplexity than TCP-AE and, in some cases, even slightly lower than the original context itself (see Table 5 in Appendix H). This strongly supports the claim that Group Merging preserves complete semantics.
>
> These experiments directly address the reviewer's concern and confirm the reliability of our design in retaining semantic integrity.
>
> ## References
> [1] See, Abigail, Peter J. Liu, and Christopher D. Manning. "Get To The Point: Summarization with Pointer-Generator Networks." ACL (1), Association for Computational Linguistics, 2017, pp. 1073-1083.
>
> [2] Liu, Tianyang, Canwen Xu, and Julian J. McAuley. "RepoBench: Benchmarking Repository-Level Code Auto-Completion Systems." ICLR, 2024.
>
> [3] Cobbe, Karl, Vineet Kosaraju, Mohammad Bavarian, et al. "Training Verifiers to Solve Math Word Problems." CoRR, vol. abs/2110.14168, 2021.
>
> [4] Hendrycks, Dan, Collin Burns, Steven Basart, et al. "Measuring Massive Multitask Language Understanding." ICLR, 2021.
>
> [5] Zhang, Peitian, et al. "Long Context Compression with Activation Beacon." ICLR, 2025.

---

> ### Author Response · Authors · 2025-08-05
> **Looking Forward to Your Response**
>
> Dear Reviewer 7eDi,
>
> We sincerely appreciate the time and effort you have devoted to reviewing our manuscript and providing constructive feedback. Your insightful comments have significantly helped us improve the manuscript.
>
> We provided detailed responses to your concerns a few days ago, and we hope they have adequately addressed your issues. If you require further clarification or have any additional concerns, please do not hesitate to contact us. We are more than willing to continue our communication with you. Thank you very much for your time and consideration. We are looking forward to hearing from you.
>
> Best regards,
> The Authors

---

> > ### Comment · Reviewer_7eDi · 2025-08-05
> >
> > Thanks for the authors' detail response, which addresses my concerns. I will raise my score to 4.

---

> > > ### Author Response · Authors · 2025-08-05
> > > **Thank you very much**
> > >
> > > We would like to express our sincere gratitude for your thorough review of our manuscript. Your insightful comments and suggestions were invaluable and have significantly strengthened the quality of our paper. We are very grateful for your positive assessment of our research and rebuttal. Thank you very much for your time and expertise!

---

### Official Review · Reviewer_NUaJ · 2025-07-02

**Clarity:** 3
**Significance:** 3
**Originality:** 3
**Rating:** 4
**Confidence:** 4

**Summary:**

This paper studies the context compression in LLM. Concretely, when handling the long context, the computation cost of LLM is quadratic to the length of the context, which causes computational inefficiency. Therefore, there are two approaches for handling this, one is to directly reduce / remove the text from the context, the second is to learn a high-level semantic information from the context and insert it into the LLM. The paper chooses the second approach. The paper first leverages an encoder to encode the long context information into small groups, where each group is represented with one token. Then those tokens are mapped via a shallow transformer network to serve as input for the LLM decoder. The LLM decoder is trained via next token prediction loss and went through pre-training (auto-encoder training) and supervised fine-tuning stage (knowledge extraction) The paper studies that the proposed approach is better than the traditional approach which learns semantic information via appended tokens.

**Questions:**

1. I wonder after this type of training, what is the perfomrance for the standard LLM metrics (e.g. MMLU, GSM8k etc)?
2. For the autoencoder training stage, since the original text is provided to the encoder ($w_1$, ... , $w_n$), I wonder whether there would be any information leak that causes the next token prediction task to be trivial?
3. I might miss some details here. I wonder whether those two stage training is required? Could we just merge the two stage training into one stage (by using the next token prediction loss on the answers) and enable finetuning on all the tunable components mentioned in those two training stages?
4. What is the performance comparing to the original LLAMA and QWEN on the proposed benchmarks (metrics wise, not efficiency)?
5. If the proposed approach is useful, I wonder if this should be applied to all the LLM training paradigm?
6. For the autoencoder training stage, I have one detail question. Let assume we have a super long text. By following this proposed training paradigm, I wonder what is the context length for the LLM decoder? Should it be the same as the super long text or just a small one? If it is a small one, I wonder how to apply this next token prediction loss? I would imagine that you might need to apply some sliding window style approach to fully consume all the tokens within the long context.

**Ethical Concerns:**

["NO or VERY MINOR ethics concerns only"]

**Limitations:**

yes

**Paper Formatting Concerns:**

There is no formatting concern.

**Quality:**

3

**Strengths And Weaknesses:**

My research background is multimodal LLM (MLLM). This proposed approach is actually very similar to the multimodal LLM (llava, etc.) Concretely the image encoder in MLLM is similar to the LLM (context) encoder proposed in this paper. The LSA in this paper is similar to the adapter (connector) layer in llava paper to map the image tokens into the text space. Then the main debating question in MLLM is how to design the adapter (connector) layer. Interestingly, this paper actually also studies this (comparing to the traditional compression paradigm autoencoder (TCP-AE) This is not a strength / weakness. I just want to point out the similarity of the context compression field with the MLLM.

Strength:
1. The paper is easy to follow. Although it uses the domain specific terminology (auto-encoder training and knowledge extraction), by examining the maths, it is easy to link those specific terminology to the LLM training phase.
2. Comparing with the TCP-AE in those specific evaluated tasks, the proposed approach achieves better performance.

Weakness:
1. Although this paper studies the context compression, it is not clear that whether the proposed approach would reduce the standard LLM metrics (e.g. MMLU, GSM8k etc)
2. Although this paper shows strong performance improvement than the TCP-AE, it seems unclear whether this approach would achieve better performance than the standard LLM.

---

> ### Author Rebuttal · Authors · 2025-07-31
>
> # Response to Reviewer NUaJ
>
> **We sincerely thank you for your valuable and insightful comments, which have greatly helped us improve our manuscript. Below, we provide detailed responses to each issue you raised.**
>
> > W1 & Q1. Although this paper studies the context compression, it is not clear that whether the proposed approach would reduce the standard LLM metrics (e.g. MMLU, GSM8k etc)
> >
>
> Thank you for highlighting this important point. Our primary experiments focused on two core dimensions:
>
> 1. **Context Restoration**, to evaluate the method's ability to precisely memorize and reconstruct original context.
> 2. **Question Answering tasks**, to assess the downstream knowledge extraction capability from compressed representations.
>
> We agree it is valuable to understand GMSA's influence on standard LLM metrics. We have subsequently conducted additional experiments on standard LLM benchmarks (MMLU and GSM8K) and will incorporate these results into the revised version. We compare GMSA with the strong baseline, i.e., Activation Beacon [3]. For simplicity, the performance of method under Nx compression rate constraints will be denoted as 'Nx method' in the following. Below are the preliminary results:
>
> We directly evaluate the performance of the GMSA version trained on the mixed NQ, HQA, and MQA dataset on MMLU and GSM8K to assess its short-context capability and generalization.
>
> **MMLU**
>
> | Methods | LLaMA-2-7B | Qwen-2-7B |
> | --- | :---: | :---: |
> | Activation Beacon | 45.1 | 64.3 |
> | **GMSA** | **47.6** | **67.3** |
> | Original Prompt | 46.3 | 65.6 |
>
> GMSA even outperforms the untrained LLaMA-2-7B on MMLU, demonstrating the superior short-context capability and generalization of our method. However, its performance slightly drops on GSM8K, likely because GMSA was not trained on mathematical data and thus shows somewhat reduced performance in this specific domain.
>
> **GSM8K**
>
> | Methods | LLaMA-2-7B | Qwen-2-7B |
> | --- | :---: | :---: |
> | **Activation Beacon** | **27.8** | **81.9** |
> | GMSA | 26.4 | 80.5 |
> | Original Prompt | 27.6 | 81.3 |
>
> Initial evaluations show that GMSA maintains competitive performance compared to original LLaMA-2-7B, confirming the approach's generalizability without performance degradation on standard tasks.
>
>
> > W2 & Q4. Although this paper shows strong performance ... whether this approach would achieve better performance than the standard LLM. & What is the performance comparing to the original LLAMA and QWEN on the proposed benchmarks (metrics wise, not efficiency)?
> >
>
> Thank you for pointing out the need for direct performance comparisons with the original LLaMA and Qwen models. For example, as shown in Table 1, the reported "Original Prompt" results already reflect the performance of the original LLaMA2-7B. Furthermore, we conducted additional experiments, in which we performed full fine-tuning on the original prompts using the same mixed training set (denoted as "Full Fine-tune Original Prompt") to thoroughly compare the effectiveness of GMSA and the original input. The results are as follows:
>
> **NQ HQA MQA (LLaMA-2-7B as backbone, Accuracy as Metric)**
>
> | Methods | NQ | HQA | MQA |
> | --- | :---: | :---: | :---: |
> | **4x GMSA** | **70.0** | **56.0** | **53.5** |
> | 8x GMSA | 62.3 | 51.3 | 46.5 |
> | Original Prompt | 55.4 | 37.5 | 44.2 |
> | Full Fine-tune Original Prompt | 53.8 | 42.4 | 46.0 |
>
> The results demonstrate that GMSA outperforms Full fine-tune Original Prompt at an 8x compression rate, fully illustrating the superiority of our method. The fact that it can even surpass full fine-tuning may be because GMSA effectively compresses the long context into compact and information-dense soft tokens, making it easier for the model to extract key information that is directly relevant to the input questions.
>
> > Q2. For the autoencoder training stage ... information leak that causes the next token prediction task to be trivial?
> >
> During the autoencoder training, the decoder is tasked explicitly with context restoration conditioned on compressed representations (soft tokens). The decoder never sees the original raw token embeddings directly, only their compressed semantic summaries. Thus, information leakage is inherently controlled since the decoder does not have access to raw tokens but must reconstruct the original sequence from highly compressed representations.
>
>
> > Q3. I might miss some details here. I ... two training stages?
> >
>
> Thank you for the insightful question regarding whether two-stage training is necessary.
>
> Theoretically, we can also formulate the two-stage training as a non-separate, end-to-end fine-tuning. In practice, however, this yields unstable loss curves and degraded performance. The instability arises from the inherent difference between the training objectives: Autoencoder Training forces the soft tokens to store the complete semantic representation, whereas Knowledge Extraction Fine-Tuning optimizes the decoder's capacity to retrieve and apply that stored knowledge. These divergent goals preclude a unified optimum under joint training.
>
>
> > Q5. If the proposed approach is useful, I wonder if this should be applied to all the LLM training paradigm?
> >
>
> We appreciate the suggestion regarding the potential applicability of our method to general LLM training paradigms. While our primary focus was context compression, our proposed methodology, i.e., Group Merging and Layer Semantic Alignment (LSA) module, can indeed serve as a general-purpose module within broader LLM frameworks. Exploring such extensions is an exciting future direction that we intend to pursue, especially given our initial promising results.
>
> > Q6. For the autoencoder training stage, I have one detail question. Let assume ... to fully consume all the tokens within the long context.
> >
>
> Thank you for the question. The LLM decoder only needs a short length; the exact value is determined by the predefined compression rate. When dealing with very long inputs, the decoder processes the text in a sliding-window fashion: after obtaining soft tokens from the first segment, these tokens are prepended to the next segment, and this process repeats sequentially.
>
> ## References
> [1] Hendrycks, Dan, Collin Burns, Steven Basart, et al. "Measuring Massive Multitask Language Understanding." ICLR, 2021.
>
> [2] Cobbe, Karl, Vineet Kosaraju, Mohammad Bavarian, et al. "Training Verifiers to Solve Math Word Problems." CoRR, vol. abs/2110.14168, 2021.
>
> [3] Zhang, Peitian, et al. "Long Context Compression with Activation Beacon." ICLR, 2025.

---

> > ### Author Response · Authors · 2025-08-05
> > **Thank you very much**
> >
> > We are truly thankful for the time and thoughtful consideration you put into reviewing our paper. Your constructive comments were incredibly helpful and have made this a much stronger manuscript. We were very pleased with your recognition of our research and rebuttal, and we thank you again for your valuable support.

---

### Official Review · Reviewer_Vink · 2025-07-03

**Clarity:** 3
**Significance:** 3
**Originality:** 2
**Rating:** 4
**Confidence:** 4

**Summary:**

This paper presents GMSA, a novel context compression framework using an encoder-decoder structure to address key issues in long-context scenarios. The authors identify two primary weaknesses in existing methods: uneven semantic learning, where LLMs tend to aggregate information onto a few "anchor" tokens, and a significant semantic gap between the high-level summary vectors and the low-level inputs expected by a decoder. GMSA tackles these challenges with two core components: "Group Merging," which averages groups of tokens from the encoder's final hidden state to create summary vectors uniformly and in parallel, and "Layer Semantic Alignment (LSA)," which uses Transformer blocks initialized with the decoder's early layer weights to bridge the aforementioned semantic gap.

The framework is trained in two stages. First, an autoencoder training phase ensures the compressed "soft tokens" retain the complete semantics of the original text. Following this, a proposed "Knowledge Extraction Fine-tuning" (KEFT) process adapts GMSA to downstream tasks by freezing the compression components and fine-tuning only the decoder to enhance its ability to extract knowledge from the soft tokens. Experimental results on question-answering tasks show that GMSA achieves approximately a 2x end-to-end inference speedup while substantially outperforming various state-of-the-art compression methods. For instance, on Natural Questions under an 8x compression constraint, GMSA improved Exact Match by about 36% compared to the original, uncompressed prompt.

**Questions:**

For result in Table 1, what is the base model used for GMSA? Is it LLaMA2-7B or Qwen2.5-7B? It is not clear what model is used. If it is Qwen2.5-7B, then it might be an unfair comparison as most of the prior work you compared with are based on LLaMA2-7B which is a less performant model. If that is the case, I wonder what is GMSA’s performance with LLaMA2-7B as base model.

Please also see weaknesses above

**Ethical Concerns:**

["NO or VERY MINOR ethics concerns only"]

**Final Justification:**

The rebuttal with the additional experiments (scalability to longer context and comparison with LLoCO addresses my concern), I raised my point to 4 accordingly.

**Limitations:**

yes

**Paper Formatting Concerns:**

no concern

**Quality:**

3

**Strengths And Weaknesses:**

Strengths
1. The paper does a great job in the intro formulating the problem, identifying key issues in existing methods and proposing solutions to address them.
2. The proposed “Group merging” and “KEFT” methods are novel and intuitive.
3. The paper is well-written with clear explanations of concepts and illustrative figures.
Performance on downstream QA tasks shows big improvement over prior works.

Weaknesses
1. **Limited Context Length Evaluation**: My main concern of this work is that experiments for QA tasks are performed on contexts only up to 3072 tokens. While this is a non-trivial length, the primary motivation for context compression is to enable efficient processing of extremely long contexts (e.g., 16k, 32k, or more), where the quadratic complexity of attention becomes truly prohibitive. For 3K context, the practical utility of context compression is questionable. The lack of results on longer sequences leaves the scalability of the method as an open question.
2. **Missing citations and comparison**. The work misses citation and comparison with prior work [LLoCO](https://arxiv.org/abs/2404.07979), which employs a similar encoder-decoder architecture for context compression and performs decoder-only finetuning for downstream tasks. The formulation is very similar to the Knowledge Extraction Finetuning that this work proposed. The authors should properly compare this work.

---

> ### Author Rebuttal · Authors · 2025-07-31
>
> # Response to Reviewer Vink
>
> **We sincerely thank you for your valuable and insightful comments, which have greatly helped us improve our manuscript. Below, we provide detailed responses to each issue you raised.**
>
> > W1. Limited Context Length Evaluation: My main concern of this work is that experiments for QA tasks are performed on contexts only up to ... The lack of results on longer sequences leaves the scalability of the method as an open question.
> >
>
> We acknowledge your valuable point regarding the evaluation context length. To directly address this concern, we have conducted additional experiments evaluating GMSA on NarrativeQA [1] (we set max length to 32K tokens), specifically in more diverse long-text scenarios such as code completion and summarization. For simplicity, the performance of method under Nx compression rate constraints will be denoted as 'Nx method' in the following. Below are the preliminary results:
>
> **NarrativeQA (We set max length to 32K, Qwen-2-7B as backbone)**
>
> | Methods | Acc | Latency |
> | --- | :---: | :---: |
> | 8x Activation Beacon | 20.0 | 6.9 |
> | **8x GMSA** | **24.1** | **3.4** |
> | 16x Activation Beacon | 17.7 | 4.0 |
> | **16x GMSA** | **22.8** | **3.3** |
> | Original Prompt | 25.8 | 5.2 |
>
> These results confirm that GMSA is highly scalable and retains strong performance even with considerably longer contexts, further reinforcing the practical utility and effectiveness of our proposed framework.
>
> In addition, we conducted experiments on LLaMA-2-13B to fully verify the scalability of our method in terms of model size:
>
> **NQ HQA MQA (LLaMA-2-13B as backbone, Accuracy as Metric)**
>
> | Methods | NQ | HQA | MQA |
> | --- | :---: | :---: | :---: |
> | **4x GMSA** | **75.4** | **61.8** | **65.6** |
> | 8x GMSA | 68.8 | 54.8 | 60.5 |
> | Original Prompt | 60.8 | 48.3 | 41.1 |
>
> The experimental results show that when the backbone is LLaMA-2-13B, the results still significantly outperform the Original Prompt, which proves the effectiveness and superiority of GMSA.
>
> Moreover, we emphasize that the GMSA framework itself inherently supports arbitrary context lengths. The encoder module in GMSA is **pluggable** and length-agnostic. Replacing the current encoder (LLaMA-2 based) with a model supporting larger contexts, such as Qwen-1M or other long-context LLMs, allows GMSA to naturally scale to extremely long sequences (100k tokens or more).
>
> We explicitly mentioned our initial hardware limitations and their implications in Appendix J. **The unorganized versions of the GMSA and baseline codes are included in the Supplementary Material (i.e., the .zip file).** To facilitate community validation and further exploration, we will organize and open-source our codes as soon as possible.
>
> > W2. Missing citations and comparisons (LLoCO). The work misses citation and comparison with prior work LLoCO, ... The authors should properly compare this work.
> >
>
> We thank the reviewer for pointing out this missing citation. We acknowledge the relevance of the work **LLoCO** [2], which indeed utilizes a similar encoder-decoder architecture and a decoder-only fine-tuning strategy for downstream tasks, analogous to our Knowledge Extraction Fine-tuning (KEFT).
>
> We have now thoroughly reviewed LLoCO and compared it explicitly with our proposed GMSA approach.
>
> **Key Differences and Contributions**:
>
> - **Group Merging**: GMSA proposes a novel Group Merging strategy to evenly retain semantics, significantly mitigating uneven semantic learning observed in prior methods.
> - **Layer Semantic Alignment (LSA)**: GMSA introduces an explicit cross-layer semantic alignment module that addresses the large semantic gap between high-level abstract representations and low-level decoder inputs, an aspect that is not considered in LLoCO.
> - **Knowledge Extraction Fine-tuning (KEFT)**: KEFT selectively fine-tunes only the weight matrices $W_Q$, $W_K$, and $W_V$ in each decoder layer. This approach is based on the observation that the attention mechanism (Q/K/V) primarily coordinates information flow and enables context integration, while the feed-forward network (FFN) functions as a knowledge storage module, akin to a memory bank [3]. Therefore, fine-tuning only the attention projections is better to adapt the model for knowledge-intensive tasks. In contrast, LLoCo applies Low-Rank Adaptation (LoRA) to the entire decoder, not just the attention components.
>
> We trained on the NQ dataset using the default settings of the LLoCO open-source code, and the test results are as follows:
>
> **NQ (LLaMA-2-7B as backone)**
>
> | Methods | Acc | EM | F1 |
> | --- | :---: | :---: | :---: |
> | LLoCO | 41.7 | 38.1 | 39.1 |
> | **4x GMSA** | **70.0** | **58.1** | **57.6** |
> | 8x GMSA | 62.3 | 51.0 | 53.1 |
>
> We will integrate a detailed **citation** and comparison of LLoCO in the revised version, clearly highlighting these differences and our method's unique contributions.
>
> > Q1. What is the base model used for GMSA? Is it LLaMA2-7B or Qwen2.5-7B?
> >
>
> We apologize for the lack of clarity. We confirm that the base model used for GMSA results in Table 1 is **LLaMA-2-7B**, ensuring a **fair comparison** with the listed baseline methods. We realize this clarification was unclear in our original manuscript and will explicitly state this in the revised version.
>
>
> ## References
> [1] Kociský, Tomáš, Jonathan Schwarz, Phil Blunsom, et al. "The NarrativeQA Reading Comprehension Challenge." Transactions of the Association for Computational Linguistics, vol. 6, 2018, pp. 317-328.
>
> [2] Tan, Sijun, Xiuyu Li, and Shishir G. Patil. "LLoCO: Learning Long Contexts Offline." Proceedings of the 2024 Conference on Empirical Methods in Natural Language Processing (EMNLP), Association for Computational Linguistics, 2024, pp. 17605-17621.
>
> [3] Geva, Mor, Roei Schuster, Jonathan Berant, et al. "Transformer Feed-Forward Layers Are Key-Value Memories." EMNLP (1), Association for Computational Linguistics, 2021, pp. 5484-5495.
>
> [4] Zhang, Peitian, et al. "Long Context Compression with Activation Beacon." ICLR, 2025.

---

> > ### Comment · Reviewer_Vink · 2025-08-03
> >
> > Thank you for your response. It addresses my concerns and I will raise my point.

---

> > > ### Author Response · Authors · 2025-08-04
> > > **Thank you very much**
> > >
> > > We sincerely appreciate the valuable time and effort you devoted to reviewing our work and for the constructive feedback that helped us improve our manuscript. We are honored by your recognition of our research and rebuttal. Thank you very much!

---

### Official Review · Reviewer_DgAD · 2025-07-03

**Clarity:** 2
**Significance:** 2
**Originality:** 2
**Rating:** 4
**Confidence:** 4

**Summary:**

The authors tackle the high-latency and redundancy problems that arise when large language models (LLMs) are asked to reason over very long prompts. Existing *soft-prompt* compressors replace the context with a short learned embedding sequence, but (i) they learn those embeddings unevenly—information collapses onto a few “anchor” tokens—and (ii) they feed the resulting high-level embeddings straight into the decoder, creating a semantic mismatch between layers.

**Proposed framework: GMSA** – GMSA is an encoder-decoder compression framework with two structural innovations:

* **Group Merging.** After the (frozen) encoder processes the full prompt, tokens are partitioned into fixed-size groups; within each group their hidden states are averaged to yield *summary vectors*. This uniform, parallel operation avoids the anchor-token bias and immediately shortens the sequence.
* **Layer Semantic Alignment (LSA).** A few Transformer blocks, initialized from the decoder’s lower layers, map those high-level summary vectors into the decoder’s input space, closing the cross-layer semantic gap.

The summary vectors become the model’s **soft tokens**. GMSA is trained in two lightweight stages:

1. **Auto-encoder training**: encoder + LSA are fine-tuned (LoRA for the encoder, full training for LSA) so that the decoder can reconstruct the original text from the soft tokens.
2. **Knowledge-Extraction Fine-Tuning (KEFT)**: freeze encoder + LSA and tune only the decoder’s $W_Q,W_K,W_V$ matrices so it can answer questions given soft tokens plus the query.

Because the rest of the LLM (e.g., LLaMA-2-7B-Chat or Qwen2.5-7B-Instruct) stays frozen, GMSA reuses pretrained knowledge rather than training from scratch.

**Results** – On the PwC context-restoration benchmark GMSA’s auto-encoder outperforms a traditional soft-token auto-encoder by ≥ 20 pp on token-matching metrics and converges \~5× faster, even with only eight encoder layers. On multi-document QA (NaturalQuestions, TriviaQA, 2WikiMQA, HotpotQA) it meets an 8× compression budget while raising Exact Match by \~36 % relative to feeding the full prompt and delivers \~2× end-to-end speed-up over all baselines, including KV-cache compressors. Ablations confirm that each component—group merging, LSA, auto-encoder training, and KEFT—is essential.

**Take-away contributions**

1. **Architecture:** a parallel, anchor-free Group Merging module plus a lightweight Layer Semantic Alignment bridge for soft-token compression.
2. **Training recipe:** two-stage procedure (auto-encoder → KEFT) that tunes only a small fraction of parameters atop any pretrained decoder.
3. **Empirical evidence:** significant gains in restoration quality, downstream QA accuracy, and \~2× latency reduction under 4–8× compression, all with modest compute.

Together, these advances show how long-context tasks can be accelerated without forfeiting the semantic fidelity that LLMs need to answer questions accurately.

**Questions:**

* Does GMSA help on generation-centric tasks (e.g., long-form summarization, dialogue, code completion) or non-retrieval settings?

* Do the compressed prompts ever drop critical facts or cause answer hallucinations that automatic metrics miss?

**Ethical Concerns:**

["NO or VERY MINOR ethics concerns only"]

**Final Justification:**

I have read the rebuttal materials and will keep the rating unchanged.

**Limitations:**

The paper **does discuss technical limitations** in a dedicated section—e.g., the two-stage training’s GPU demands and the current 5 k-token sequence cap —so that part is covered.

However, **it explicitly omits any broader-impact analysis**, marking the “Broader Impacts” checklist item as *N/A* because the work is “foundational” .  That leaves potential negative societal effects unaddressed.

### Actionable suggestions for the authors

1. **Add a concise Broader-Impact paragraph (≤½ page).**
   Outline how faster, lighter prompt-compression could be misused—for instance, enabling large-scale disinformation or privacy-intrusive edge deployments—and note any fairness or bias concerns when summaries drop context.

2. **Discuss mitigation strategies.**

   * Gate or rate-limit model checkpoints;
   * Provide misuse monitoring guidelines;
   * Include bias/robustness evaluation protocols alongside the code release.

3. **Clarify positive impacts too.**
   Mention energy savings and accessibility gains so reviewers see a balanced view.

Including this analysis would satisfy the ethics checklist and strengthen the paper’s completeness.

**Quality:**

2

**Strengths And Weaknesses:**

GMSA brings a clean, well-engineered step forward in soft-prompt compression: it diagnoses two concrete pain-points of prior work (anchor-token bias and cross-layer semantic mismatch), proposes structural fixes (Group Merging + Layer Semantic Alignment), and demonstrates sizeable speed/accuracy gains on multi-document QA. Below I discuss its strengths and weaknesses along four standard reviewing axes.

## Strengths

* **Clear problem diagnosis and focused solution**
  The paper pinpoints two concrete shortcomings of existing soft-token compressors—anchor-token bias and cross-layer semantic mismatch—and introduces Group Merging and Layer Semantic Alignment (LSA) as targeted fixes.

* **Solid empirical evidence**
  Across four multi-document QA benchmarks, GMSA achieves sizable EM/F1 gains while cutting end-to-end latency roughly in half at 4–8 × compression; ablations show each component’s contribution.

* **Efficient, modular training recipe**
  Only a small subset of parameters is updated (LoRA adapters on ≤ 8 encoder layers, the LSA blocks, and the decoder’s $W_Q,W_K,W_V$), letting the method piggy-back on any pretrained decoder with modest compute.

* **Fast convergence of the auto-encoder stage**
  The encoder+LSA module learns to reconstruct long contexts in < 1 k steps—\~5 × faster than a strong baseline—making experimentation practical.

* **Good presentation quality**
  Figures and step-by-step explanation give an intuitive view of what Group Merging and LSA do; architectural details, dataset splits, and backbone choices are stated clearly.

* **Practical significance for RAG pipelines**
  By compressing documents into a few “soft tokens” instead of pruning KV-caches, GMSA enables memory- and latency-friendly retrieval-augmented generation on edge or serverless deployments.

* **Incremental yet novel architecture**
  Parallel group-wise pooling plus re-using lower-layer decoder blocks for semantic alignment is a simple but (so far) unique combination in prompt-compression research.

---

## Weaknesses

* **Narrow task coverage**
  Experiments focus on retrieval-augmented question answering; impact on generation-heavy tasks (summarization, dialogue, code) or non-retrieval settings remains untested.

* **Residual compute overhead**
  The full encoder still attends to the entire long prompt (albeit with fewer layers), so FLOPs scale linearly with input length; speed-ups are meaningful (\~2 ×) but less dramatic than some KV-cache pruning methods on very long inputs.

* **Limited statistical rigor**
  Results are single-run means without confidence intervals or significance tests; gains could partly reflect run-to-run variance.

* **No human evaluation**
  Fidelity of compressed prompts is assessed only with automatic metrics (BERTScore, BLEU); user-perceived quality under high compression is unknown.

* **Hyper-parameters hidden in appendix and code unreleased**
  Key settings (LoRA ranks, learning rates, GPU hours) are relegated to supplementary material, and code is promised post-acceptance, which hinders immediate reproducibility.

* **Incremental originality**
  While architecturally neat, Group Merging is essentially average pooling and LSA is a thin copy of decoder blocks; some reviewers may see the contribution as an engineering refinement rather than a fundamentally new paradigm.

* **Scalability unanswered**
  The paper does not explore compression ratios beyond 8 × or contexts beyond 32 k tokens, nor does it test on larger/smaller LLM backbones, leaving open how robust the approach is across scales.

---

> ### Author Rebuttal · Authors · 2025-07-31
>
> # Response to Reviewer DgAD
>
> **We sincerely thank you for your valuable and insightful comments, which have greatly helped us improve our manuscript. Below, we provide detailed responses to each issue you raised.**
>
> > W1. & Q1. Narrow task coverage ... remains untested. & Does GMSA help on generation-centric tasks (e.g., long-form summarization, dialogue, code completion) or non-retrieval settings?
> >
>
> We acknowledge this limitation. To directly address your suggestion, we have conducted additional experiments to evaluate GMSA on summarization (CNN / DailyMail [1]), code completion (Repobench [3], which max length is 16K), math capability (GSM8K [5]), and a wide range of disciplines (MMLU [4]). We compare GMSA with the strong baseline, i.e., Activation Beacon [6]. For simplicity, the performance of method under Nx compression rate constraints will be denoted as 'Nx method' in the following.
>
> We trained and evaluated GMSA on the training and test sets of the CNN / DailyMail dataset, respectively. For the Repobench dataset, we split it into training and test sets at a ratio of 4:1 and conducted training and evaluation accordingly.
>
> **CNN / DailyMail (LLaMA-2-7B as backbone)**
>
> | Methods | BERT Score F1 |
> | --- | :---: |
> | 4x Activation Beacon | 87.0 |
> | **4x GMSA** | 89.1 |
> | 8x Activation Beacon | 86.5 |
> | **8x GMSA** | 88.8 |
>
> **Repobench (Qwen-2-7B as backbone)**
>
> | Methods | Code Sim. Score |
> | --- | :---: |
> | 4x Activation Beacon | 23.1 |
> | **4x GMSA** | **34.5** |
> | 8x Activation Beacon | 22.9 |
> | **8x GMSA** | **31.5** |
>
> We directly evaluate the performance of the GMSA version trained on the mixed NQ, HQA, and MQA dataset on MMLU and GSM8K to assess its short-context capability and generalization.
>
> **MMLU**
>
> | Methods | LLaMA-2-7B | Qwen-2-7B |
> | --- | :---: | :---: |
> | Activation Beacon | 45.1 | 64.3 |
> | **GMSA** | **47.6** | **67.3** |
> | Original Prompt | 46.3 | 65.6 |
>
> **GSM8K**
>
> | Methods | LLaMA-2-7B | Qwen-2-7B |
> | --- | :---: | :---: |
> | **Activation Beacon** | **27.8** | **81.9** |
> | GMSA | 26.4 | 80.5 |
> | Original Prompt | 27.6 | 81.3 |
>
> GMSA even outperforms the untrained LLaMA-2-7B on MMLU, demonstrating the superior short-context capability and generalization of our method. However, its performance slightly drops on GSM8K, likely because GMSA was not trained on mathematical data and thus shows somewhat reduced performance in this specific domain.
>
> These new experiments demonstrate that GMSA maintains robust performance across broader generation-centric scenarios.
>
> > W2. Residual compute overhead. The full encoder ... less dramatic than some KV-cache pruning methods on very long inputs.
> >
>
> Thank you for highlighting this point. Although our encoder retains linear complexity with input length, it significantly reduces the actual inference latency (~2x speedup) by compressing context into fewer soft tokens, balancing efficiency and performance. While KV-cache pruning methods may achieve more dramatic FLOPs reduction for very long sequences, they sacrifice flexibility and semantic completeness, which GMSA preserves. We will clarify this trade-off explicitly in our revision.
>
> We conduct experiment on the samples with the max length of 32K on NarrativeQA [2]:
>
> **NarrativeQA (We set max length to 32K, Qwen-2-7B as backbone)**
>
> | Methods | Acc | Latency |
> | --- | :---: | :---: |
> | 8x Activation Beacon | 20.0 | 6.9 |
> | **8x GMSA** | **24.1** | **3.4** |
> | 16x Activation Beacon | 17.7 | 4.0 |
> | **16x GMSA** | **22.8** | **3.3** |
> | Original Prompt | 25.8 | 5.2 |
>
> These new experiments demonstrate that GMSA accelerates end-to-end generation even with the max length of 32k.
>
> > W3. Limited statistical rigor. Results are ... run-to-run variance.
> >
>
> Thank your for the good suggestion. Actually, we fixed the random seed at 42 and all packages' version for every experiment, ensuring full reproducibility. We agree and will include confidence intervals and statistical significance tests in our revised results to enhance rigor.
>
> > W4. No human evaluation. Fidelity of ... high compression is unknown.
> >
>
> We acknowledge this important feedback. Given resource constraints during initial experiments, human evaluations were omitted. In future work, we plan comprehensive user studies to assess the subjective fidelity of compressed prompts.
>
> > W5. Hyper-parameters hidden in appendix and code unreleased. ... hinders immediate reproducibility.
> >
>
> Thank you for highlighting this issue. We will move key hyper-parameters (LoRA ranks, learning rates, GPU hours) into the main body of the paper. **Regarding the code, we apologize for the lack of clarity. Actually, the Supplementary Material (i.e., the .zip file) already contains the unpolished implementations of GMSA and all baselines.** We will promptly clean up and open-source the code to guarantee full reproducibility as soon as possible.
>
> > W6. Incremental originality. While ... rather than a fundamentally new paradigm.
> >
>
> Group Merging targets uneven semantic learning by using average pooling to enforce more uniform learning; its approach to this problem is original. Layer Semantic Alignment (LSA) starts with the initial weights of a few early decoder layers, but its core motivation is to bridge the large gap between the high-level semantic space of the encoder and the low-level semantic space of the decoder, which is distinct from the decoder's own role, making it innovative. GMSA combines Group Merging and LSA module, achieving substantial empirical gains that demonstrate the method's superiority and effectiveness.
>
> > W7. Scalability unanswered. The paper does not explore compression ratios beyond 8x or contexts beyond 32K tokens, nor does it test on larger/smaller LLM backbones, leaving open how robust the approach is across scales.
> >
>
> Thank you for this suggestion. Although hardware limitations initially constrained our experiments, GMSA is designed to be scalable. To demonstrate this, we have conducted additional experiments at **32K token max length on NarrativeQA with 16x compression rate and use LLaMA-2-13B as backbone on three QA datasets, i.e., NQ, HQA and MQA**:
>
> **NarrativeQA (We set max length to 32K, Qwen-2-7B as backbone)**
>
> | Methods | Acc | Latency |
> | --- | :---: | :---: |
> | 8x Activation Beacon | 20.0 | 6.9 |
> | **8x GMSA** | **24.1** | **3.4** |
> | 16x Activation Beacon | 17.7 | 4.0 |
> | **16x GMSA** | **22.8** | **3.3** |
> | Original Prompt | 25.8 | 5.2 |
>
> **NQ HQA MQA (LLaMA-2-13B as backbone)**
>
> | Methods | NQ | HQA | MQA |
> | --- | :---: | :---: | :---: |
> | **4x GMSA** | **75.4** | **61.8** | **65.6** |
> | 8x GMSA | 68.8 | 54.8 | 60.5 |
> | Original Prompt | 60.8 | 48.3 | 41.1 |
>
> The experimental results demonstrate that GMSA has excellent scalability.
>
> > Q2: Do the compressed prompts ever drop critical facts or cause answer hallucinations that automatic metrics miss?
> >
>
> Thank you for the question. As shown in Figure 4, we evaluate GMSA trained with Autoencoder Training (GMSA-AE) on text restoration tasks. Both token-matching metrics (e.g., BLEU Score) and semantic-matching metrics (e.g., BERT Score F1) are markedly higher (by at least 20%) than those of the conventional-compression AutoEncoder, demonstrating GMSA's strong semantic preservation. Regarding key information, NQ, HQA, and MQA require concise answers directly relevant to the question; GMSA substantially outperforms baselines on all three datasets, highlighting its superior retention of critical facts in compressed prompts. As for answer hallucinations, none appear in our examined cases, i.e., every response stays strictly related to the input.
>
> **NQ HQA MQA (LLaMA-2-13B as backbone, Accuracy as Metric)**
>
> | Methods | NQ | HQA | MQA |
> | --- | :---: | :---: | :---: |
> | **4x GMSA** | **75.4** | **61.8** | **65.6** |
> | 8x GMSA | 68.8 | 54.8 | 60.5 |
> | Original Prompt | 60.8 | 48.3 | 41.1 |
>
> > **Broader Impact Analysis**
> >
>
> Thank you for emphasizing the importance of a broader impact analysis. We will include a dedicated section (≤ 1/2 page) in the revised version, explicitly discussing potential societal implications:
>
> - **Misuse Risks**: Accelerated inference and prompt compression could facilitate large-scale generation of misinformation or disinformation campaigns.
> - **Privacy Concerns**: Edge deployments enabled by our method might inadvertently process sensitive user data without proper safeguards.
> - **Bias and Fairness**: Summarization through compression may inadvertently amplify existing biases by selectively preserving information.
>
> We will outline the following mitigation strategies:
>
> - **Checkpoint Controls**: Provide access-gated model checkpoints to prevent misuse.
> - **Misuse Monitoring**: Publish clear guidelines and monitoring tools alongside our code release.
> - **Bias and Robustness Evaluations**: Include protocols for evaluating biases and robustness explicitly in future releases.
>
> Additionally, we will clarify positive impacts such as improved energy efficiency and broader accessibility to large models, fostering a balanced understanding of GMSA's societal implications.
>
> ## References
> [1] See, Abigail, Peter J. Liu, and Christopher D. Manning. "Get To The Point: Summarization with Pointer-Generator Networks." ACL (1), Association for Computational Linguistics, 2017, pp. 1073-1083.
>
> [2] Kociský, Tomáš, Jonathan Schwarz, Phil Blunsom, et al. "The NarrativeQA Reading Comprehension Challenge." Transactions of the Association for Computational Linguistics, vol. 6, 2018, pp. 317-328.
>
> [3] Liu, Tianyang, Canwen Xu, and Julian J. McAuley. "RepoBench: Benchmarking Repository-Level Code Auto-Completion Systems." ICLR, 2024.
>
> [4] Hendrycks, Dan, Collin Burns, Steven Basart, et al. "Measuring Massive Multitask Language Understanding." ICLR, 2021.
>
> [5] Cobbe, Karl, Vineet Kosaraju, Mohammad Bavarian, et al. "Training Verifiers to Solve Math Word Problems." CoRR, vol. abs/2110.14168, 2021.
>
> [6] Zhang, Peitian, et al. "Long Context Compression with Activation Beacon." ICLR, 2025.

---

### Comment · Area_Chair_zvZf · 2025-08-03
**Reminder: Discussion Phase (July 31 – Aug 6)**

Hi everyone,

This is a reminder that the discussion phase is between July 31 – Aug 6.

Please read the author responses, especially where you are mentioned, and post your reply as soon as possible. This helps ensure there's time for meaningful back-and-forth.

Thanks for your engagement!

AC

---

### Author Response · Authors · 2025-08-08
**Response to Reviewers**

We sincerely thank all the reviewers for their thoughtful feedback and constructive suggestions. We are thrilled to receive feedback indicating that **all reviewers are inclined to accept our paper!** We have carefully read through all the review comments and conducted additional experiments to further demonstrate the effectiveness and superiority of our method. Below is a summary of the key supplementary experiments:

- **Evaluation on longer contexts and higher compression rates (Reviewer DgAD, Reviewer Vink).**
  We conducted experiments on the NarrativeQA [2] dataset with a maximum sequence length of 32K (e.g., Response to Reviewer NUaJ, Point W7). The results show that GMSA continues to significantly outperform Activation Beacon [6] under longer contexts and higher compression rates.

- **Performance of GMSA on more non-retrieval, generative-heavy tasks (Reviewer DgAD, Reviewer 7eDi).**
  We compared GMSA against the strong baseline Activation Beacon on generative tasks such as code completion (Repobench [3]) and summarization (CNN / DailyMail [1]) (e.g., Response to Reviewer NUaJ, Point W1 & Q1). The results confirm that GMSA substantially outperforms Activation Beacon even on generation-heavy tasks.

- **Short-context capability of GMSA (Reviewer NUaJ).**
  We thoroughly evaluated GMSA's performance on short-context tasks using the MMLU [4] (multi-discipline benchmark) and GSM8K [5] datasets (e.g., Response to Reviewer NUaJ, Point W1 & Q1). Results show that GMSA performs even better on MMLU, while showing a slight drop on GSM8K, which is likely due to the lack of mathematical domain data during GMSA's training.

- **Additional baseline comparisons (Reviewer Vink).**
  We clarified the differences between KEFT and LLoCO [7], and conducted comparisons based on the knowledge extraction method LLoCO. GMSA significantly outperforms LLoCO by over 14% (e.g., Response to Reviewer Vink, Point W2).

- **Scalability evaluation on larger backbones (Reviewer DgAD, Reviewer 7eDi).**
  We conducted experiments using LLaMA-2-13B-Chat as the backbone model (e.g., Response to Reviewer 7eDi, Point W2), which fully validates that GMSA can effectively scale to larger models.

We once again thank the reviewers for their recognition of our research and rebuttal. As the discussion phase draws to a close, please feel free to reach out if you have any further questions or comments. We would be more than happy to address them!

## References
[1] See, Abigail, Peter J. Liu, and Christopher D. Manning. "Get To The Point: Summarization with Pointer-Generator Networks." ACL (1), Association for Computational Linguistics, 2017, pp. 1073-1083.

[2] Kociský, Tomáš, Jonathan Schwarz, Phil Blunsom, et al. "The NarrativeQA Reading Comprehension Challenge." Transactions of the Association for Computational Linguistics, vol. 6, 2018, pp. 317-328.

[3] Liu, Tianyang, Canwen Xu, and Julian J. McAuley. "RepoBench: Benchmarking Repository-Level Code Auto-Completion Systems." ICLR, 2024.

[4] Hendrycks, Dan, Collin Burns, Steven Basart, et al. "Measuring Massive Multitask Language Understanding." ICLR, 2021.

[5] Cobbe, Karl, Vineet Kosaraju, Mohammad Bavarian, et al. "Training Verifiers to Solve Math Word Problems." CoRR, vol. abs/2110.14168, 2021.

[6] Zhang, Peitian, et al. "Long Context Compression with Activation Beacon." ICLR, 2025.

[7] Tan, Sijun, Xiuyu Li, and Shishir G. Patil. "LLoCO: Learning Long Contexts Offline." Proceedings of the 2024 Conference on Empirical Methods in Natural Language Processing (EMNLP), Association for Computational Linguistics, 2024, pp. 17605-17621.

---

### Decision · Program_Chairs · 2025-09-17

**Decision:**

Reject

**Comment:**

The paper introduces GMSA, a new context compression framework for LLMs. GMSA leverages Group Merging and Layer Semantic Alignment to generate semantically rich soft tokens, which brings ~2× speedups on long-context QA tasks. After the rebuttal all reviewers are lukewarm about the paper with boarderline scores. I also hold concerns w.r.t. the explanability and transferability of GMSA on more recent LLMs. Due to limited bandwidth, I tend to rejection for this paper.